# FCVL: FOURIER CROSS-VIEW LEARNING FOR GENERALIZABLE 3D OBJECT DETECTION IN BIRD'S EYE VIEW

## ABSTRACT

Improving the generalization of Bird's Eye View (BEV) detection models is essential for safe driving in the real world. In this paper, we consider a realistic yet more challenging scenario, which aims to improve the generalization with single source data for training, as collecting multiple source data is time-consuming and labor intensive in autonomous driving. To this end, we rethink the task from a frequency perspective and exploit the cross-view consistency between adjacent perspectives. We propose the Fourier Cross-View Learning (FCVL) framework including Fourier Hierarchical Augmentation (FHiAug), an augmentation strategy in the frequency domain to boost domain diversity and Fourier Cross-View Semantic Consistency Loss to facilitate the model to learn more domain-invariant features. Furthermore, we provide theoretical guarantees via augmentation graph theory. To the best of our knowledge, this is the first study to explore generalizable 3D Object Detection in BEV with single-source data. Extensive experiments on various testing domains have demonstrated that our approach achieves the best performance on various test domains with single-source data.

## 1 INTRODUCTION

Recent advances in Bird's Eye View (BEV) representations have shown significant potential for multi-camera 3D object detection, as they capture both spatial locations and semantic features without being heavily affected by occlusions. While existing camera-based BEV models (Philion & Fidler, 2020; Huang et al., 2022; Li et al., 2022b;c) have achieved excellent performance on in-distribution datasets like nuScenes (Caesar et al., 2020), they struggle in real-world settings where the environment and conditions vary widely. This performance drop occurs because camera data in practical applications often has different distributions compared to the limited training data. As a result, enhancing the generalization of these models is critical for their safe deployment. Domain generalization (DG) aims to generalize a model to an unseen target domain by learning from multiple source domains. However, collecting diverse source data for training is time-consuming and labor-intensive, especially in autonomous driving scenarios, and cannot always guarantee improved performance. In this paper, we tackle a more practical yet challenging problem: improving the generalization of 3D object detectors when trained on a single source domain. Focusing on single-domain generalization (SDG) not only addresses practical constraints but also provides a more robust evaluation of model adaptability.

In SDG for 2D image classification, previous works (Zhao et al., 2023; Qiao et al., 2020) aim to enhance data diversity using common 2D data augmentation techniques[1], such as geometric transformations, style transfer, or adversarial data generation. However, directly applying these approaches to BEV-based tasks introduces several challenges. First, BEV representations are generated by projecting multi-view 2D features using real-world physical constraints, which limits the use of strong geometric transformations, such as 270-degree rotations, as they would disrupt the spatial consistency of the BEV space. Second, style transfer techniques (Zhao et al., 2023) replace the original image statistics with those from the target style, but this often blurs the boundary between style and content (Lee et al., 2023), distorting important features and ultimately harming model generalization. Third, adversarial generation methods(Goodfellow et al., 2020) suffer from unstable training and mode

---

[1]We have provided a more detailed introduction to these techniques in the Appendix A.

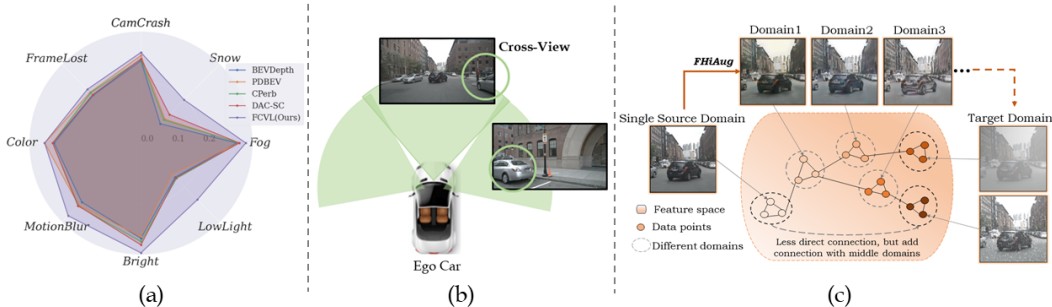

Figure 1: (a) Detection results of different models: the proposed *FCVL* can improve the generalization of 3D detection on multiple target domains with single source training data. (b) Cross-View Learning: make the most of the natural cross-view input to improve the generalization. (c) Augmentation graph connectivity: augmentations of data from the same classes are assumed to be connected. FHiAug increases the augmentation graph connectivity between source and unseen domains.

collapse. While diffusion-based techniques(Ho et al., 2020) are more stable, they add significant computational and storage overhead, making them impractical for complex 3D detection models. Therefore, common 2D data augmentations cannot be effectively leveraged to create diverse training samples for BEV-based tasks. More importantly, for multi-camera 3D object detection, the natural availability of multi-view data offers a unique opportunity to learn domain-invariant features, a potential that remains underexplored in scenarios with limited training data.

In response to these limitations and challenges, we propose the Fourier Cross-View Learning (FCVL) framework including Fourier Hierarchical Augmentation (FHiAug), an augmentation strategy in the frequency domain to boost domain diversity and Fourier Cross-View Semantic Consistency Loss to facilitate the model to learn more domain-invariant features. Different from Zhao et al. (2023) expanding style statistics in the pixel domain, we utilize the Fourier transform to introduce style variations while minimizing content distortion. This is motivated by the well-known property of the Fourier transformation: the phase component encodes high-level semantic information, while the amplitude component captures low-level image statistics (Xu et al., 2021). This separation allows us to independently manipulate style (low-level statistics) and content (high-level semantics) in the frequency domain. At the image level, we introduce Frequency Jitter, which perturbs both amplitude and phase components to create diverse samples that complement the single source domain. At the feature level, we propose Amplitude Transfer, a novel method for generating fine-grained style variations, ensuring domain diversity in the latent space. For multi-camera setups, FHiAug applies cross-camera augmentation, creating surrounding views with varied "styles" to simulate realistic variations. To leverage the natural multi-view input, we design the Fourier Cross-View Semantic Consistency Loss, which aligns adjacent perspectives to help the model develop robust features against domain shifts. Furthermore, using augmentation graph theory (HaoChen et al., 2022; Wang et al., 2024), we provide a unique theoretical perspective on FCVL and establish its theoretical soundness.

In summary, our major contributions are as follows:

- Towards SDG for multi-camera 3D object detection, we present the Fourier Cross-View Learning framework to fully exploit natural cross-view inputs.

- We propose FHiAug, a novel, efficient, plug-and-play augmentation strategy that operates on both image and feature levels, to enhance domain diversity without requiring additional modules or specialized training strategies.

- We propose Fourier Cross-View Semantic Consistency Loss to facilitate the model to learn more domain-invariant features from adjacent perspectives.

- Using augmentation graph theory, we provide a valid theoretical foundation for the effectiveness of FCVL.

- To the best of our knowledge, this is the first work to address generalizable 3D object detection in BEV using single-source data. Extensive experiments across various test domains demonstrate that our approach achieves superior performance compared to existing domain generalization methods (See Fig.1(a)).

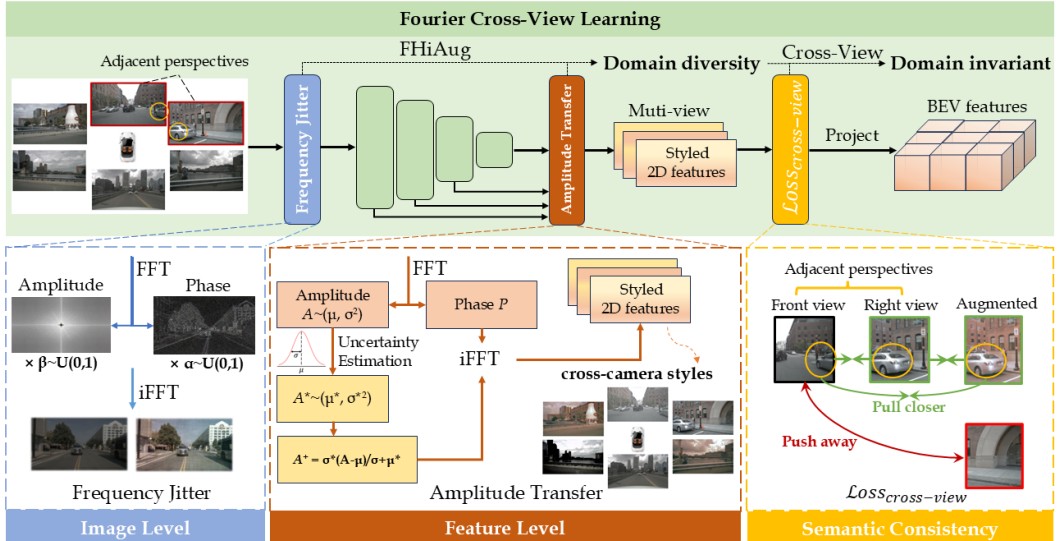

Figure 2: Overview of our FCVL framework. FCVL includes two major parts: FHiAug to boost domain diversity and Fourier Cross-View Semantic Consistency Loss to ensure domain-invariant BEV features. FHiAug consists of two stages. One is Frequency Jitter at image level. The other is Amplitude Transfer at feature level. Notably, we achieve cross-camera augmentation via FHiAug, which means a set of surrounding views have different "styles". This forces the model to learn from diversified domains. Besides, on multi-view features, we calculate Fourier Cross-View Semantic Consistency Loss to learn more domain-invariant BEV features.

## 2 METHODOLOGY

### 2.1 OVERVIEW OF FOURIER CROSS-VIEW LEARNING (FCVL) FRAMEWORK

In this section, we elaborate the Fourier Cross-View Learning (FCVL) framework. The FCVL framework is motivated by the cross-view consistency in BEV 3D object detection. For example, as shown in Fig.1(b), objects such as cars or pedestrians are often visible across multiple adjacent camera views. This overlap results in similar BEV features across these cameras, providing an inductive bias to guide the learning process. To capture this cross-view relationship, we implement a Fourier Cross-View Semantic Consistency Loss, where features from nearby camera views are considered positive samples, while those from distant views are treated as negative samples. To enhance feature diversity and improve domain generalization, we propose Fourier Hierarchical Augmentation, which applies frequency-based transformations to different camera views. This method enriches the feature space, promoting the learning of domain-invariant BEV features. The overall structure of our framework is depicted in Fig.2 and the process is outlined in Algorithm 1 in Appendix D. In the following sections, we provide an in-depth explanation of both the Fourier Hierarchical Augmentation and the Fourier Cross-View Semantic Consistency Loss.

### 2.2 FOURIER HIERARCHICAL AUGMENTATION

Fourier Hierarchical Augmentation (FHiAug) includes data augmentation at image level (Frequency Jitter) and domain perturbation at feature level (Amplitude Transfer), which is a plug-and-play and non-parameter method to boost domain diversity without extra module designing or special training strategies.

#### 2.2.1 FREQUENCY JITTER AT IMAGE LEVEL

For a single channel image $x \in \mathcal{R}^{d_1 \times d_2}$, the 2D Fourier transformation is defined as follows,

$$F(x)(u,v) = \sum_{m=0}^{d_1-1} \sum_{n=0}^{d_2-1} x(m,n) \exp^{-2\pi i(\frac{mu}{d_1} + \frac{nv}{d_2})}, \qquad (1)$$

where $F$ denotes Fourier Transform; $u$ and $v$ denote spatial coordinates; $m$ and $n$ denote frequency coordinates.

The amplitude components $\mathcal{A}$ and phase components $\mathcal{P}$ are then respectively expressed as:

$$\mathcal{A}(x)(u,v) = [R^2(x)(u,v) + I^2(x)(u,v)]^{1/2}, \mathcal{P}(x)(u,v) = \arctan\left[\frac{I(x)(u,v)}{R(x)(u,v)}\right], \quad (2)$$

where $R(x)$ and $I(x)$ represent the real and imaginary part of $F(x)$, respectively.

To generate diverse samples that complement the single source domain, we employ two strategies. First, we perturb the amplitude component using a hyperparameter, $\alpha$, to create variations in low-level statistics. Second, we modify the intensity of the phase component with a hyperparameter, $\beta$, to expose the model to previously less emphasized features (Chen et al., 2020).

$$\hat{\mathcal{A}}(x)(u,v) = \alpha\mathcal{A}(x)(u,v), \hat{\mathcal{P}}(x)(u,v) = \beta\mathcal{P}(x)(u,v), \quad (3)$$

where $\alpha \sim U(\eta, 1)$ and the hyperparameter $\eta$ control the strength of the augmentation on amplitude; $\beta \sim U(\lambda, 1)$ and the hyperparameter $\lambda$ control the strength of the augmentation on phase.

With new amplitude and phase component, we can form a new Fourier representation and use inverse Fourier transformation to generate the augmented image $\hat{x}$.

$$F(\hat{x})(u,v) = \hat{\mathcal{A}}(x)(u,v) * e^{-j*\hat{\mathcal{P}}(x)(u,v)}, \hat{x} = F^{-1}[F(\hat{x})(u,v)]. \quad (4)$$

In the training phase, we set $p_i$ as the calling probability of Frequency Jitter and sample $p \sim U(0,1)$. For image input $x$, we acquire the augmented $x_{aug}$ as:

$$x_{aug} = \text{Frequency\_Jitter}(x), \text{if } p \le p_i. \quad (5)$$

This Fourier-based augmentation strategy, termed Frequency Jitter, manipulates both amplitude and phase components, as shown in Fig.6. The top row demonstrates adjustments to the amplitude, primarily affecting image brightness, which helps the model become robust to varying lighting conditions. The bottom row shows modifications to the phase component, creating samples with varying levels of semantic detail while preserving the overall structure. This controlled manipulation of semantic strength encourages the model to learn more domain-invariant and robust features. Additional examples highlighting the effect of phase adjustments are provided in Fig.8(b).

### 2.2.2 AMPLITUDE TRANSFER AT FEATURE LEVEL

To implement domain perturbation and create diverse virtual styles during training, we apply Amplitude Transfer based on the style statistics of intermediate features. This approach aims to improve model robustness and generalization.

Given an intermediate feature map $\mathbf{X} \in \mathcal{R}^{B \times C \times H \times W}$, where $B$, $C$, $H$, and $W$ denote batch size, number of channels, height, and width, respectively, we first perform a Fourier transformation and extract its amplitude component $\mathcal{A}(\mathbf{X}) \in \mathcal{R}^{B \times C \times H \times W}$. We then compute the channel-wise mean ($\mu$) and standard deviation ($\sigma$) for each instance's amplitude as follows:

$$\mu(\mathcal{A}(\mathbf{X})) = \frac{1}{HW}\sum_{h=1}^{H}\sum_{w=1}^{W}\mathcal{A}(\mathbf{X}), \sigma^2(\mathcal{A}(\mathbf{X})) = \frac{1}{HW}\sum_{h=1}^{H}\sum_{w=1}^{W}[\mathcal{A}(\mathbf{X}) - \mu(\mathcal{A}(\mathbf{X}))]^2. \quad (6)$$

Now, we acquire the style statistics $\mu(\mathcal{A}(\mathbf{X}))$ and $\sigma^2(\mathcal{A}(\mathbf{X}))$ of the features. To achieve feature-level perturbation, different from Xu et al. (2021) and Zhou et al. (2021) to mix up different domains' style information directly, inspired by Li et al. (2022a) we make uncertainty estimation on $\mu(\mathcal{A}(\mathbf{X}))$ and $\sigma(\mathcal{A}(\mathbf{X}))$ with the variance as follows:

$$\text{Var}(\mu(\mathcal{A}(\mathbf{X}))) = \frac{1}{B}\sum_{b=1}^{B}[\mu(\mathcal{A}(\mathbf{X})) - \mathbb{E}(\mu(\mathcal{A}(\mathbf{X})))]^2,$$

$$\text{Var}(\sigma(\mathcal{A}(\mathbf{X}))) = \frac{1}{B}\sum_{b=1}^{B}[\sigma(\mathcal{A}(\mathbf{X})) - \mathbb{E}(\sigma(\mathcal{A}(\mathbf{X})))]^2, \quad (7)$$

where $B$ is the batch size and $\mathbb{E}$ denotes the mathematical expectations.

Next, we obtain new style statistics $\beta$ and $\gamma$ by random sampling from the Gaussian distributions:

$$\beta(\mathcal{A}(\mathbf{X})) = \mu(\mathcal{A}(\mathbf{X})) + \epsilon_\mu \sqrt{\text{Var}(\mu(\mathcal{A}(\mathbf{X})))}, \epsilon_\mu \sim \mathcal{N}(0,1),$$
$$\gamma(\mathcal{A}(\mathbf{X})) = \sigma(\mathcal{A}(\mathbf{X})) + \epsilon_\sigma \sqrt{\text{Var}(\sigma(\mathcal{A}(\mathbf{X})))}, \epsilon_\sigma \sim \mathcal{N}(0,1).$$

(8)

Finally, we replace the original style statistics with the perturbed values and perform an inverse Fourier transform to obtain the augmented feature map $\hat{\mathbf{X}}$:

$$\hat{\mathcal{A}}(\mathbf{X}) = \gamma(\mathcal{A}(\mathbf{X})) \times \frac{\mathcal{A}(\mathbf{X}) - \mu(\mathcal{A}(\mathbf{X}))}{\sigma(\mathcal{A}(\mathbf{X}))} + \beta(\mathcal{A}(\mathbf{X})).$$

(9)

This allows us to create diverse styled features in each training iteration without explicitly defining content and style. During training, we set $p_f$ as the probability of applying Amplitude Transfer and sample $p \sim U(0,1)$. For a given feature input $\mathbf{X}$, the augmented feature $\mathbf{X}_{aug}$ is generated as follows:

$$\mathbf{X}_{aug} = \text{Amplitude\_Transfer}(\mathbf{X}), \text{if } p \leq p_f.$$

(10)

We visualize the style variations of some pictures via Amplitude Transfer in Fig.7. The left column shows the original images, while the adjacent columns display styled variations. As observed, the augmented images exhibit different colors and textures, showcasing the effectiveness of the proposed method in generating diverse feature styles.

### 2.3 FOURIER CROSS-VIEW SEMANTIC CONSISTENCY LOSS

For multi-camera 3D object detection, the input inherently includes cross-view data, which is beneficial for learning domain-invariant features. This has not yet been harnessed to improve generalization. As illustrated in Fig.3, consider a car appearing in both the front and front-right views. Such cross-view targets are common in multi-camera inputs, providing natural opportunities to observe the same object from different perspectives. To exploit this, we propose the Fourier Cross-View Semantic Consistency Loss to help the model learn more robust features from adjacent views. Unlike conventional consistency losses that operate in the pixel domain, we minimize the distance between the *phase* distributions of the targets with the same semantics, as the *phase* component usually en-

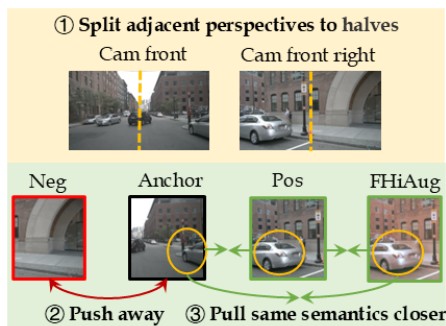

Figure 3: Illustration of Fourier Cross-View Semantic Consistency Loss

codes high-level semantic information. Concretely, for adjacent views, we split the features into halves as shown in Fig.3. We treat the target from the right half of the first view as the anchor, use the same target or the augmented one via FHiAug from the left half of the adjacent view as the positive sample and select other samples as negatives. Next, we calculate triplet loss (Schroff et al., 2015) in the frequency domain to explore potential semantic similarity as follows:

$$\begin{cases} \text{view}^{\text{pos}}_{\text{aug}} &= \text{FHiAug}(\text{view}^{\text{pos}}), \\ \text{view}^{\text{neg}}_{\text{aug}} &= \text{FHiAug}(\text{view}^{\text{neg}}), \end{cases}$$

(11)

$$a = \mathcal{P}(\text{view}^{\text{anchor}}), p = \mathcal{P}(\text{view}^{\text{pos}}_{\text{aug}}), n = \mathcal{P}(\text{view}^{\text{neg}}_{\text{aug}}),$$

(12)

$$\mathcal{L}_{\text{cross}} = \max(\text{dist}(a,p) - \text{dist}(a,n) + \text{margin}, 0),$$

(13)

where FHiAug is the proposed augmentation method; $\text{view}^{\text{anchor}}$ is the anchor example; $\text{view}^{\text{pos}}$ is the sample with the same category as anchor; $\text{view}^{\text{neg}}$ is the sample with different categories; $\mathcal{P}$ denotes calculating the phase components of different views after Fourier transformation; dist is the distance measurement; margin is a constant greater than zero.

Overall, the training objective loss including detection loss $\mathcal{L}_{\text{det}}$ and consistency loss $\mathcal{L}_{\text{cross}}$ can be written as:

$$\mathcal{L}_{\text{train}} = \mathcal{L}_{\text{det}} + \gamma \mathcal{L}_{\text{cross}},$$

(14)

where $\gamma$ is the weighting parameter to balance different loss terms.

# 3 THEORETICAL ANALYSIS

To analyze the influence of data augmentation, we adopt the standard augmentation graph framework (HaoChen et al., 2022; Wang et al., 2024), where data augmentations induce interactions (as edges) between training samples (as nodes). Given a natural data sample $\overline{x} \in \overline{\mathcal{X}}$, we use $\mathcal{A}(\cdot|\overline{x})$ to denote the distribution of its augmentations. For any two augmented data $x, x' \in \mathcal{X}$, define the adjacency matrix $W_{xx'}$ as the marginal probability of $x$ and $x'$ from a random natural data $\overline{x} \sim \mathcal{P}_{\overline{\mathcal{X}}}$:

$$W_{xx'} = \mathbb{E}_{\overline{x} \sim \mathcal{P}_{\overline{\mathcal{X}}}}[\mathcal{A}(x|\overline{x})\mathcal{A}(x'|\overline{x})]. \tag{15}$$

Let $\mathbb{L} = I - D^{-\frac{1}{2}}WD^{-\frac{1}{2}}$ be the normalized graph Laplacian matrix, where $D$ is a diagonal degree matrix with the $(x, x)$-th diagonal element as $D_{xx} = \sum_{x'} W_{xx'}$.

Based on the above augmentation graph framework, we construct the augmentation graph $G(\mathcal{X}, \overline{\mathcal{X}}, W)$ in the feature space for single source domain and augmented domains as shown in Fig.1(c). Then, we have the following theorem:

**Theorem 1.** *For the optimal encoder $f^*$, BEV projection module $P^*_{BEV}$, a learned classification head $C^*$ and regression head $R^*$ on augmented data $\mathcal{X}$, its linear probing error has the following generalization upper bound,*

$$\mathcal{E}(f^*, P^*_{BEV}, C^*, R^*) \leq \frac{2\alpha}{\lambda_{k+1}} + 4\Delta(y_c, \hat{y}_c) + 4\Delta(y_r, \hat{y}_r), \tag{16}$$

*where $\alpha$ denotes the labeling error caused by data augmentation; $\lambda_{k+1}$ denotes the $k + 1$-th smallest eigenvalue of the Laplacian matrix $\mathbb{L}$; $\Delta(y_c, \hat{y}_c)$ denotes the average disagreement between $\hat{y}_c$ and the ground-truth labeling $y_c$ for classification; $\Delta(y_r, \hat{y}_r)$ denotes the average disagreement between $\hat{y}_r$ and the ground-truth labeling $y_r$ for regression.*

Based on the generalization upper bound in Eq.16 (proof in AppendixE.1), we can provide rigorous explanations to show that our method can increase graph connectivity $\lambda_{k+1}$ and reduce label error $\alpha$ to decrease the generalization loss.

First, as shown in Fig.1(c), as we can only get access to the single source data, the connectivity of the graph is poor and only a few feature points are connected. There is a large margin between the source and target domains. The proposed FHiAug plays a positive role in expanding graph connectivity $\lambda_{k+1}$, since it creates more diverse "middle" domains between single source data and unseen target domains. According to augmentation graph theory, with the increase of augmentation strength, the graph connectivity $\lambda_{k+1}$ can be increased. Via increasing $\lambda_{k+1}$ in Eq.16, the generalization upper bound can be tighter and the generalization ability can be improved.

However, common strong augmentation, such as strong geometric enhancement, also causes label error (larger $\alpha$ in Eq.16 ) and increases the generalization loss. The proposed FHiAug augmenting in the frequency domain can effectively alleviate this issue. Next, we will provide a theoretical analysis and show that FHiAug can ensure semantic consistency under strong augmentation strength to increase connectivity. As mentioned in Sec.2, input data $\mathcal{X}$ can be decomposed into two components: phase $\mathcal{X}_p$, and amplitude $\mathcal{X}_a$, where $\mathcal{X}_p$ contains semantic information about the label $y$, denoting the causal component, and $\mathcal{X}_a$ contains more low-level information, denoting the non-causal components.

**Assumption 1.** *We assume the linear relationship between $\mathcal{X}_p$ and $y$,*

$$y = \mathcal{X}_p \phi + \epsilon, \tag{17}$$

*where $\epsilon$ is the noise, $Cov(\mathcal{X}_p, \epsilon) = 0$, $\mathbf{E}[\mathcal{X}_p] = 0$.*

**Theorem 2.** *If input data $\mathcal{X}$ consists of all the phase components, $\mathcal{X} = \mathcal{X}_p$, the optimal linear predictor $\phi$ can be estimated without bias. Otherwise, the predictor $\phi$ is biased.*

For some style transferring methods in pixel domain, both phase and amplitude components are modified. In this situation, the predictor $\phi$ is biased, which means that the predictor probably gives wrong prediction of label. While the proposed method FHiAug augments in the frequency domain and retains the phase congruency, avoiding label error effectively. At image level, Frequency Jitter only adjusts the intensity of phase component in the global. The distribution of semantic information is not changed. At feature level, we achieve style transfer with operating on amplitude component only. More proof for Theorem 2 is in Appendix E.2.

Table 1: Comparison with baseline methods on nuScenes and nuScenes-C. The table represents the results of **NDS** ↑ with ResNet50 as backbone. "Clean" denotes the normal validation set of nuScenes. "OoD Avg." is the average performance of eight testing domains. FCVL achieves **SOTA** out-of-distribution performance on three frameworks. PD-BEV †(Lu et al., 2023) has released the code for BEVDepth. Thus, we mainly compare our method with PDBEV on BEVDepth for fair comparison. The best and second-best results are highlighted in Red and Blue, respectively.

| Model | Clean | Cam Crash | Frame Lost | Color Quant | Motion Blur | Bright | Low Light | Fog | Snow | OoD Avg. |
|---|---|---|---|---|---|---|---|---|---|---|
| BEVFormer (Li et al., 2022c) | 0.4362 | 0.3175 | 0.3246 | 0.3410 | 0.2549 | 0.4022 | 0.2461 | 0.3853 | 0.1510 | 0.3028 |
| +CPerb (Zhao et al., 2023) | 0.4356 | 0.3199 | 0.3292 | 0.3372 | 0.2548 | 0.4096 | 0.2420 | 0.3907 | 0.1661 | 0.3062 |
| +DSU (Li et al., 2022a) | 0.4359 | 0.3206 | 0.3322 | 0.3609 | 0.3425 | 0.4083 | 0.2458 | 0.3937 | 0.2601 | 0.3330 |
| +DAC-SC (Lee et al., 2023) | 0.4332 | 0.3085 | 0.2872 | 0.3703 | 0.3691 | 0.4161 | 0.3155 | 0.4093 | 0.3086 | 0.3481 |
| +FACT (Xu et al., 2021) | 0.4379 | 0.3181 | 0.3285 | 0.3436 | 0.2585 | 0.4100 | 0.2494 | 0.3916 | 0.1486 | 0.3060 |
| +FCVL(Ours) | 0.4375 | 0.3244 | 0.3374 | 0.3751 | 0.3748 | 0.4202 | 0.3078 | 0.4170 | 0.2969 | 0.3567 |
| BEVDepth (Li et al., 2022b) | 0.4028 | 0.2654 | 0.2178 | 0.2801 | 0.2697 | 0.3072 | 0.1558 | 0.3080 | 0.0881 | 0.2365 |
| PD-BEV † (Lu et al., 2023) | 0.4094 | 0.2822 | 0.2316 | 0.3102 | 0.2842 | 0.3011 | 0.1411 | 0.3151 | 0.1091 | 0.2468 |
| +CPerb (Zhao et al., 2023) | 0.4034 | 0.2698 | 0.2294 | 0.2847 | 0.2873 | 0.3180 | 0.1616 | 0.3164 | 0.1054 | 0.2466 |
| +DSU (Li et al., 2022a) | 0.4057 | 0.2722 | 0.2330 | 0.3065 | 0.3270 | 0.3462 | 0.2165 | 0.3249 | 0.1565 | 0.2729 |
| +DAC-SC (Lee et al., 2023) | 0.4007 | 0.2714 | 0.2200 | 0.2846 | 0.2861 | 0.3284 | 0.1586 | 0.3172 | 0.1299 | 0.2495 |
| +FACT (Xu et al., 2021) | 0.4026 | 0.2670 | 0.2224 | 0.2872 | 0.2749 | 0.3276 | 0.1611 | 0.3141 | 0.0957 | 0.2438 |
| +FCVL(Ours) | 0.4050 | 0.2722 | 0.2346 | 0.3106 | 0.3318 | 0.3539 | 0.2577 | 0.3380 | 0.1968 | 0.2870 |
| BEVDet (Huang et al., 2022) | 0.3880 | 0.2508 | 0.1955 | 0.2409 | 0.2201 | 0.2591 | 0.1112 | 0.2633 | 0.0728 | 0.2017 |
| +CPerb (Zhao et al., 2023) | 0.3908 | 0.2590 | 0.2065 | 0.2479 | 0.2325 | 0.2643 | 0.1322 | 0.2752 | 0.0782 | 0.2120 |
| +DSU (Li et al., 2022a) | 0.3835 | 0.2582 | 0.2061 | 0.2814 | 0.3019 | 0.3128 | 0.1806 | 0.2961 | 0.1065 | 0.2430 |
| +DAC-SC (Lee et al., 2023) | 0.3884 | 0.2574 | 0.2046 | 0.2688 | 0.2644 | 0.2986 | 0.1450 | 0.2926 | 0.1028 | 0.2293 |
| +FACT (Xu et al., 2021) | 0.3907 | 0.2581 | 0.2054 | 0.2430 | 0.2277 | 0.2708 | 0.1230 | 0.2727 | 0.0773 | 0.2098 |
| +FCVL(Ours) | 0.3848 | 0.2579 | 0.2064 | 0.2928 | 0.3204 | 0.3244 | 0.2393 | 0.3156 | 0.1848 | 0.2677 |

# 4 EXPERIMENTS

## 4.1 EXPERIMENTS SETUP

To verify different methods' generalization ability, we first utilize nuScenes (Caesar et al., 2020) as the single training source and nuScenes-C (Xie et al., 2023) as the testing sets. To further demonstrate the effectiveness of our method, we experiment on Argoverse 2 (Wilson et al., 2023). We choose four baselines including BEVFormer, BEVDepth, BEVDet and new SOTA method Far3D (Jiang et al., 2023a). More details of datasets and implementation can be found in Appendix B.

## 4.2 COMPARISON WITH SOTA METHODS

We compare our method with some SOTA SDG and DG methods which involve frequency-domain data augmentation (CPerb (Zhao et al., 2023), FACT (Xu et al., 2021)) and style transformation (DAC-SC (Lee et al., 2023)). Besides, PD-BEV (Lu et al., 2023) working on BEVDepth, is proposed to ensure consistent and accurate detection and improve generalization via perspective debiasing.

The results on nuScenes and nuScenes-C are shown in Table 1. **On transformer-based framework**, our FCVL can greatly improve the generalization of BEVFormer as shown in Table 1. The average NDS of eight testing domains is increasing from 0.3028 to 0.3567 (↑ 5.39%). FCVL achieves SOTA out-of-domain performance across different SDG or DG methods. **On LLS-based framework**, FCVL achieves SOTA performance on both BEVDepth and BEVDet as well. In terms of the average NDS of eight testing domains, our method achieves much better performance than BEVDepth (↑ 5.05%) and BEVDet (↑ 6.07%). Especially, FCVL improves the performance of BEVDepth by 10.87% for adverse weather conditions *Snow* and 10.19% for *Low Light*. Similarly, FCVL improves the performance of BEVDet by 8.07% for adverse weather conditions *Snow* and 12.50% for *Low Light*. FCVL has more stable generalization ability for adverse weather and light conditions on different 3D detection frameworks. For **worst cases** *Low Light* and *Snow*, as shown in Fig.4(b), FCVL has shown significant improvement. **Overall**, as is shown in Fig.4(a), the proposed FCVL outperforms other methods with great margin on average of three frameworks (↑ 2.08%). Besides, for both transformer-based framework and LLS-based framework, FCVL has the superiority in stable maintenance of better generalization ability in eight testing domains, especially in *Low Light*, *Motion Blur* and *Snow*.

To comprehensively evaluate our method, we extend our methods to the 3D detectors without explicit BEV features, such as Sparse4D(Lin et al., 2023) and multi-modal method, such as BEVFusion(Liu et al., 2024) as well. We list the experimental results including different 3D detection schemes (explicit BEV or not, multi frames or not, etc.) in the Table 3. Our method can improve the out-of-distribution performance in all the settings, while maintaining the in-distribution performances.

More results on Argoverse 2 are shown in Table 2. We experiment on a new SOTA Far3D, which presents a sparse query-based method for multi-view 3D long-range detection without explicit BEV features. To achieve training on one domain and test on unseen domains, we sample data from sunny weather in urban scenarios as the training data and data from cloudy weather or city scenarios as the ood test set. As is shown, our method improves the generalization for long-range detection as well.

Table 2: Comparison with baseline methods on Argoverse 2. The table represents the results of **mAP** ↑. "Clean" denotes the in-domain set.

| Model | Clean | City | Cloudy | OoD Avg. |
|---|---|---|---|---|
| Far3D(Jiang et al., 2023a) | 0.219 | 0.146 | 0.113 | 0.130 |
| +CPerb (Zhao et al., 2023) | 0.221 | 0.156 | 0.130 | 0.143 |
| +DSU (Li et al., 2022a) | 0.218 | 0.168 | 0.138 | 0.153 |
| +DAC-SC (Lee et al., 2023) | 0.213 | 0.162 | 0.140 | 0.151 |
| +FACT (Xu et al., 2021) | 0.220 | 0.159 | 0.129 | 0.144 |
| +FCVL(Ours) | 0.220 | 0.176 | 0.161 | **0.169** |

Table 3: The table represents the effectiveness of our proposed method under different settings on nuScenes and nuScenes-C. "C" denotes camera input. "L" denotes lidar input. "Explicit BEV" means 3D detectors generate explicit BEV features. "Temporal" denotes whether utilizing multi frames. "Depth" denotes whether utilizing depth information. Bold fonts indicate the best results.

Table 4: Ablation Study on different components of FCVL on BEVDepth(Li et al., 2022b). Amplitude means only jittering on amplitude component. Phase means only jittering on phase component. Jittering on both is the Frequency Jitter operated at image level. AT denotes Amplitude Transfer at feature level. Bold fonts indicate the best results.

| Model | Modality | Temporal | Depth | Explicit BEV | Clean | OoD Avg. |
|---|---|---|---|---|---|---|
| BEVFormer | C | ✓ | | ✓ | 0.4362 | 0.3028 |
| +FCVL(Ours) | C | ✓ | | ✓ | **0.4375** | **0.3567** |
| BEVDepth | C | | ✓ | ✓ | 0.4028 | 0.2365 |
| +FCVL(Ours) | C | | ✓ | ✓ | **0.4050** | **0.2870** |
| BEVDepth | C | ✓ | ✓ | ✓ | 0.4828 | 0.4128 |
| +FCVL(Ours) | C | ✓ | ✓ | ✓ | 0.4827 | **0.4291** |
| BEVDet | C | | | ✓ | 0.3880 | 0.2017 |
| +FCVL(Ours) | C | | | ✓ | 0.3848 | **0.2677** |
| Sparse4Dv3 | C | ✓ | ✓ | | 0.5590 | 0.4431 |
| +FCVL(Ours) | C | ✓ | ✓ | | **0.5592** | **0.4492** |
| BEVFusion | L+C | | | ✓ | 0.7074 | 0.6865 |
| +FCVL(Ours) | L+C | | | ✓ | **0.7123** | **0.6948** |

| Amplitude | Phase | AT | $\mathcal{L}_{cross}$ | Clean | OoD Avg. |
|---|---|---|---|---|---|
| | | | | 0.4028 | 0.2365 |
| ✓ | | | | 0.4037 | 0.2735 |
| | ✓ | | | 0.4021 | 0.2690 |
| ✓ | ✓ | | | 0.4037 | 0.2767 |
| | | ✓ | | 0.4022 | 0.2570 |
| ✓ | ✓ | ✓ | | 0.4004 | 0.2843 |
| ✓ | ✓ | ✓ | ✓ | **0.4050** | **0.2870** |

## 4.3 ABLATION STUDY ON NUSCENES

### 4.3.1 EFFECTS OF DIFFERENT COMPONENTS OF FCVL

Firstly, we analyze the effects of different components of Frequency Jitter at image level, as shown in Table 4. On average, jittering phase of the input only or jittering amplitude only has impressive performance. When jittering both phase and amplitude, Frequency Jitter improves the performance further. Then, we combine Frequency Jitter and Amplitude Transfer to further improve all the testing domains' performance, which can demonstrate that the strength of proposed augmentations at image and feature levels. At last, we add our $\mathcal{L}_{cross}$ in the training. Notably, the consistency loss not only is beneficial for the in-domain performance, but also boosts the out-of-domain performance.

### 4.3.2 EFFECTS OF DIFFERENT INSERTED POSITIONS OF AMPLITUDE TRANSFER

We evaluate the impact of different inserted positions of Amplitude Transfer, as shown in Table 8. Inserted position of ResNet is numbered as follows: after first Conv 0, after Max Pooling layer 1, after first Resblock 2, after second Resblock 3, after third Resblock 4 and after fourth Resblock 5, respectively. According to Zhou et al. (2021), Resblock 1 to Resblock 3 contain domain-related information, which means domain-related information usually lies in shallow layers. Thus, in our method, Amplitude Transfer is inserted in Position 0-2. We make more experiments by increasing Position 3-5 gradually to find more suitable positions. As shown in Table 8, in terms of in- and out- of distribution performance, inserting Amplitude Transfer in Position 0-3 achieves both the best performance.

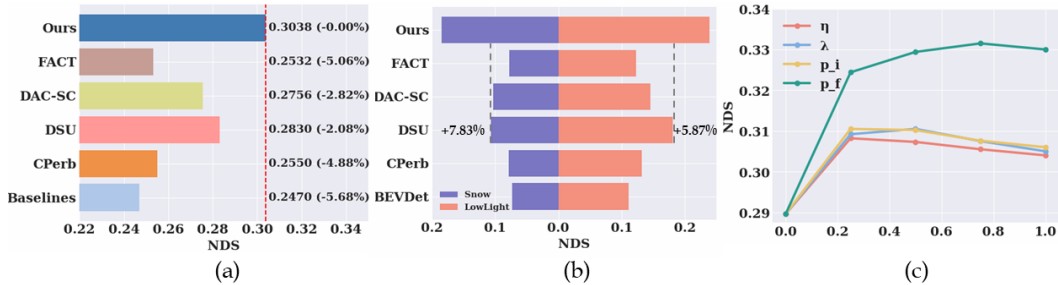

(a)          (b)          (c)

Figure 4: (a) The average detection results of different methods including eight OoD domains under three baseline frameworks. As is shown, the proposed FCVL outperforms other methods with great margin on average. (b) Worst cases analysis. Our method has shown significant improvement in the worst cases, *Low Light* and *Snow*. (c) Hyperparameters analysis of FHiAug. The strength of augmentation $\eta$ and $\lambda$ for Frequency Jitter; the probability $p_i$ for Frequency Jitter and $p_f$ for Amplitude Transfer.

### 4.3.3 EFFECTS OF HYPERPARAMETERS

In Frequency Jitter, there are three hyper-parameters. The hyperparameter $\eta$ controls the strength of Amplitude augmentation; $\lambda$ controls the strength of Phase augmentation and $p_i$ is the provability of implementing Frequency Jitter. For Amplitude Transfer, as we have decided where to insert AT in above section, in this section, we experiment on the probability $p_f$ of implementing Amplitude Transfer. As shown in Fig.4(c), initially, as the probability and intensity increase, the out-of-domain performance gradually improves. After reaching a certain level of probability and intensity, further changes in the parameters will no longer cause drastic changes in ood performance, indicating that the model is stable against hyper-parameter misspecifications as long as the hyper-parameters are within reasonable ranges. We set $\eta = 0.25$, $\lambda = 0.5$, $p_i = 0.25$ and $p_f = 0.75$ as the final setting.

Furthermore, we conduct experiments to validate the effect of Fourier Cross-View Semantic Consistency Loss on BEVDet. We take Fourier Cross-View Semantic Consistency Loss as a stand alone addition to the backbone and adjust the weights of $\mathcal{L}_{\text{cross}}$. The results are shown in the Table 9. $\gamma$ is the weight of $\mathcal{L}_{\text{cross}}$. It can be seen that when adding this Cross-View Semantic Consistency Loss separately, the overall generalization performance has been significantly improved, especially in some domains such as Motion Blur, LowLight.

### 4.4 EFFICIENCY ANALYSIS

In this part, we make efficiency analysis to delve into the proposed FCVL. We investigate how the method scales with increasing image resolution and computational complexity. The results are listed in Table 5. With larger resolution, FCVL can still improve the performance. FCVL is only used during the training phase. In the inference, we do not need to do the augmentation. Our approach enhances the algorithm's generalization performance without increasing the time consumption during the inference, which is beneficial for practical applications.

Table 5: Efficiency analysis of FCVL. Training time refers to the time it takes for one training step when the batch size is 1. Inference time refers to the time for inferring a single sample. "Memory" is the consumed GPU memory during training with batch size 1. All the tests are conducted on RTX 3090 GPU.

| Model | Resolution | Training time (s) | Inference time (s) | Memory(MB) | OoD Avg. |
|-------|-----------|-------------------|---------------------|------------|----------|
| BEVDet | $256 \times 704$ | 0.257 | **0.073** | 5498 | 0.2017 |
| +FCVL | $256 \times 704$ | 0.364 | **0.073** | 7383 | 0.2677($\uparrow 6.60\%$) |
| BEVDet | $512 \times 1408$ | 0.482 | **0.143** | 11698 | 0.2006 |
| +FCVL | $512 \times 1408$ | 0.605 | **0.143** | 20094 | 0.2394($\uparrow 3.88\%$) |

### 4.5 VISUALIZATION ANALYSIS

We use t-SNE to visualize the BEV features from different domains of BEVDet and FCVL. In the Fig. 5, source domain is represented in red and other colors represent different target domains. We can find that the features of different domains extracted from BEVDet are distant from each other and loosely distributed in the feature space.

While, after optimization with FCVL, the distribution of four domains becomes more compact and connected, which is in line with augmentation graph theory. FCVL increases the augmentation graph connectivity between source and unseen domains and improve the generalization a lot. More visualized results of FCVL can be found in Appendix F.

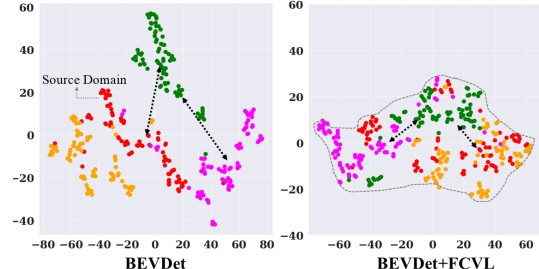

Figure 5: t-SNE Visualization of FCVL.

## 5 RELATED WORK

### 5.1 MULTI-VIEW 3D OBJECT DETECTION

The recent advances in BEV representation exhibit great potential for multi-view 3D Object Detection Dong et al. (2024); Yang et al. (2023); Pan et al. (2024); Qi et al. (2024); Li et al. (2024); Zhang et al. (2022); Jiang et al. (2023b). The camera-based BEV models (Philion & Fidler, 2020; Huang et al., 2022; Li et al., 2022b;c) have achieved excellent performance on in-distribution datasets but improving the generalization of such detection models in real-world application scenarios is remains under-studied. PD-BEV renders diverse view maps from BEV features and rectify the perspective bias of these maps to help the learning of features resilient to domain shifts (Lu et al., 2023). DG-BEV creates multiple pseudo-domains and construct an adversarial training loss to encourage the feature representation to be more domain-agnostic (Wang et al., 2023a).

### 5.2 SINGLE DOMAIN GENERALIZATION

Domain Generalization (DG) aims to generalize a model trained on multiple source domains to a target domain which is distributionally different. CIRL(Lv et al., 2022) generates augmented images by a causal intervention module with intervention upon non-causal factors. AGFA (Kim et al., 2023) trains the classifier and the amplitude generator adversarially to synthesise a worst-case domain for adaptation. This paper focuses on single domain generalization (Wang et al., 2023b) which is a more challenging yet realistic setting. Wang et al. (2023b) propose a style-complement module to enhance the generalization power of the model by synthesizing images from diverse distributions that are complementary to the source ones. Chen et al. (2023) propose a new learning paradigm, namely simulate-analyze-reduce, which first simulates the domain shift by building an auxiliary domain as the target domain, then learns to analyze the causes of domain shift, and finally learns to reduce the domain shift for model adaptation. Qiao et al. (2020) leverage adversarial training to create "fictitious" yet "challenging" populations and use a Wasserstein Auto-Encoder (WAE) to relax the widely used worst-case constraint in a meta-learning scheme. Zhao et al. (2023) propose CPerb, a simple yet effective cross-perturbation method to enhance the diversity of the training data and introduce multi-route perturbation to learn domain-invariant features. As can be seen from previous work, increasing data diversity is a key ingredient for single domain generalization.

## 6 CONCLUSION

In conclusion, this paper addresses the challenge of Single Domain Generalization in multi-camera 3D object detection via Fourier Cross-View Learning framework. We propose a non-parametric Fourier Hierarchical Augmentation at both image and feature levels to enhance data diversity and Fourier Cross-View Semantic Consistency Loss to facilitate model to learn more domain-invariant features from adjacent perspectives. Besides, via augmentation graph theory, we make valid theoretical guarantees. Extensive experiments on various testing domains of different datasets have demonstrated that our approach achieves the best performance across various domain generalization methods.

**Limitations.** At present, there are several hyperparameters to be tuned. In the future work, we can explore additional techniques to avoid spending too much time on tuning hyperparameters. Additionally, for snowy weather, we have already improved by 10 points, but the performance is still much worse compared to the performance in other scenarios such as low light. Consequently, there is a substantial potential for enhancement in adverse weather conditions.

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

# A    INTRODUCTION TO COMMON 2D DATA AUGMENTATION TECHNIQUES

In SDG for 2D image classification, previous works (Zhao et al., 2023; Qiao et al., 2020) aim to enhance data diversity using common 2D data augmentation techniques, such as geometric transformations, style transfer, or data generation. However, directly applying these approaches to BEV-based tasks introduces several challenges.

First, BEV representations are generated by projecting multi-view 2D features using real-world physical constraints, which limits the use of strong geometric transformations, such as 270-degree rotations, as they would disrupt the spatial consistency of the BEV space. We add a strong geometric enhancement experiment including significant rotation and translation on the image and the results are as follows in the Table 6. For one thing, it shows that strong geometric enhancement hurts in-domain performance, as large scale rotation or translation may destroy physical restraints in the real driving scenario. For another, geometric enhancement is not very effective in improving OoD performance.

Table 6: Strong geometric enhancement experiments including significant rotation and translation on the images.

| Model | Clean | Cam Crash | Frame Lost | Color Quant | Motion Blur | Bright | Low Light | Fog | Snow | OoD Avg. |
|---|---|---|---|---|---|---|---|---|---|---|
| BEVDet | 0.3880 | 0.2508 | 0.1955 | 0.2409 | 0.2201 | 0.2591 | 0.1112 | 0.2633 | 0.0728 | 0.2017 |
| strong geo | 0.3505 | 0.2338 | 0.1875 | 0.2249 | 0.2030 | 0.2371 | 0.1188 | 0.2511 | 0.0639 | 0.1900 |

Second, some style transfer techniques (Zhao et al., 2023; Nuriel et al., 2021) replace the original image statistics with those from the target style, but this often blurs the boundary between style and content (Lee et al., 2023), distorting important features and ultimately harming model generalization. These methods need to remove the "style" in the pixel domain first. Some content cues will be removed inevitably.

Third, data generation methods including adversarial generation and diffusion-based techniques. Training a Generative Adversarial Network (GAN) (Goodfellow et al., 2020) involves a competitive process between two neural networks: the generator and the discriminator. Adversarial generation can suffer from unstable training and mode collapse. It often requires a lot of experiments and fine-tuning to get a GAN to work well. While diffusion-based techniques(Ho et al., 2020) are more stable, they need significant computational and storage overhead, making them impractical for complex 3D detection models. Additionally, although we can spend much time generating a large number of samples, we would also require extra storage space. However, our method involves online augmentation and does not require any additional storage space.

Therefore, common 2D data augmentations cannot be effectively leveraged to create diverse training samples for BEV-based tasks.

Besides, we further clarify the differences between our method and other frequency-domain approaches. Compared with these methods(Xu et al., 2021; Lv et al., 2022; Kim et al., 2023), our method has major strengths in two aspects including accuracy and efficiency. Firstly, in the setting of single source data, our proposed method can enhance the generalization ability of the detectors by large margin. FACT (Xu et al., 2021) mixes up two different domains' data in frequency, e.g. Cartoon and Photo from dataset PACS and achieves great OOD performance in the paper. But when training with only single domain, FACT can only mix the samples within the single domain and it indeed improves the in-domain clean set a little, but the improvement of OOD sets is very slim in the single domain setting. Different from FACT, we first propose Frequency Jitter at image level to create diverse samples that are complementary to the single source domain. Then, at feature level, we introduce a novel method Amplitude Transfer to achieve style transfer without content distortion. Through uncertainty estimation, we can obtain uncertain feature statistics, which can gradually shift the features to more diverse domains through continuous training.

Secondly, due to the high complexity of BEV-based 3D object detection models, our plug-and-play and non-paramerter data augmentation method can achieve better generalization results more efficiently. CIRL (Lv et al., 2022) generates augmented images by a causal intervention module with intervention upon non-causal factors. AGFA (Kim et al., 2023) trains the classifier and the amplitude generator adversarially to synthesise a worst-case domain for adaptation. Compared with

these methods, our proposed method is simple, stable, yet effective without extra module designing or special training strategies.

# B   MORE DETAILS OF EXPERIMENTS SETUP

## B.1   DATASETS

To verify different methods' single domain generalization ability, we first utilize nuScenes (Caesar et al., 2020) as the single training source and nuScenes-C (Xie et al., 2023) as the testing sets. NuScenes-C is comprehensive dataset that encompasses eight distinct corruptions, including Bright, Dark, Fog, Snow, Motion Blur, Color Quant, Camera Crash, and Frame Lost. Each type of corruption has three different levels of corruption intensity (i.e., easy, moderate, and hard). These eight corruptions include different weather conditions, different light conditions, potential equipment damage situations. These scenarios are common out-of-distribution problems in real-world application. We use eight distinct corruptions as our multi testing domains to evaluate the effectiveness of different DG methods.

Besides, we experiment on public dataset Argoverse 2 (Wilson et al., 2023), which includes different scenarios from different cities. We sample 20 % data to achieve single domain training set and evaluate five categories including 'BICYCLE', 'LARGE VEHICLE', 'MOTORCYCLE', 'PEDESTRIAN', 'REGULAR VEHICLE'.

## B.2   EVALUATION METRIC.

For 3D detection task, we maily report mean Average Precision (mAP) and nuScenes Detection Score (NDS) (Caesar et al., 2020), which is calculated of mAP, as well as five True Positive (TP) metrics including mean Average Translation Error (mATE), mean Average Scale Error (mASE), mean Average Orientation Error (mAOE), mean Average Velocity Error (mAVE), mean Average Attribute Error (mAAE).

$$\text{NDS} = \frac{1}{10}[5\text{mAP} + \sum_{\text{mTP} \in \mathbb{TP}} (1 - \min(1, \text{mTP}))] \tag{18}$$

where, $\mathbb{TP}$ is the set of the five mean True Positive metrics.

## B.3   IMPLEMENTATION DETAILS

To comprehensively evaluate the performance of generalization algorithms, we first choose three baselines BEVFormer, BEVDepth and BEVDet and experiment on nuScenes. We use ResNet50 as the backbone for these baselines. We extend our method on three baselines respectively. All parameters in our framework are initialized from ImageNet. We apply an AdamW optimizer with the learning rate set to 0.0002 and we set the batch size to 2 per GPU. All experiments are conducted with 4 3090 GPUs.

Besides, we choose a new SOTA Far3D (Jiang et al., 2023a) as another baseline and experiment on dataset Argoverse 2.

# C   MORE EXPERIMENTAL RESULTS

## C.1   EXPERIMENTS WITH RANDOM SEED

In early experiments, we find that the effect of random seeds on BEVDepth or BEVFormer is relatively small. We run our method FCVL three times on BEVDepth and the average NDS on clean testing set is $0.4004 \pm 0.0002$; the average NDS of OoD sets is $0.2845 \pm 0.0003$. The standard deviation for three trials is 0.0002 or 0.0003, which means the method is quite robust to different seeds. Thus, in later experiments, we run our method with random seed.

## C.2 EVALUATION IN PRACTICAL APPLICATION SCENARIOS

To further evaluate the performance of our algorithm in practical application scenarios, we collect a large dataset consisting of sunny daytime and nighttime. We train the detection model with our method on 61716 samples of sunny daytime and test on daytime (6169) and night (8200) samples. More results can be found in Table 7. Notably, on the night testing set, FCVL can improve the mAP from 0.0420 to 0.1004($\uparrow$ 5.84%).

Table 7: Evaluation results (**mAP** $\uparrow$) in the real-world autonomous driving scenarios.

| Model | Daytime | Night |
|---|---|---|
| Baseline | 0.2690 | 0.0420 |
| +FCVL(Ours) | 0.2687 | 0.1004($\uparrow$ 5.84%) |

## C.3 EXTRA ABLATION STUDY RESULTS ON NUSCENES

Effects of different inserted positions of Amplitude Transfer are shown in Table 8. The effect of Fourier Cross-View Semantic Consistency Loss alone are shown in Table 9.

Table 8: Effects of different inserted positions of Amplitude Transfer. Inserted position of ResNet is numbered as: after first Conv 0, after Max Pooling layer 1, after first Resblock 2, after second Resblock 3, after third Resblock 4 and after fourth Resblock 5. "0-5" means inserting Amplitude Transfer from Position 0 to Position 5.

| Model | Clean | OoD Avg. |
|---|---|---|
| BEVFormer | 0.4362 | 0.3028 |
| 0-5 | 0.4404 | 0.3267 |
| 0-4 | 0.4393 | 0.3289 |
| 0-3 | **0.4421** | **0.3294** |
| 0-2 | 0.4404 | 0.3280 |

Table 9: The effect of Fourier Cross-View Semantic Consistency Loss alone. $\gamma$ is the weight of $\mathcal{L}_{\mathrm{cross}}$.

| Model | Cam Crash | Frame Lost | Color Quant | Motion Blur | Bright | Low Light | Fog | Snow | OoD Avg. |
|---|---|---|---|---|---|---|---|---|---|
| BEVDet | 0.2508 | 0.1955 | 0.2409 | 0.2201 | 0.2591 | 0.1112 | 0.2633 | 0.0728 | 0.2017 |
| $\gamma = 0.5$ | 0.2487 | 0.1942 | 0.2444 | 0.2132 | 0.2583 | 0.1328 | 0.2635 | 0.0641 | 0.2024 |
| $\gamma = \mathbf{1.0}$ | 0.2501 | 0.1952 | 0.2785 | 0.2882 | 0.2890 | 0.1407 | 0.2807 | 0.1140 | **0.2296** |
| $\gamma = 2.0$ | 0.2462 | 0.1932 | 0.2806 | 0.2863 | 0.2872 | 0.1340 | 0.2803 | 0.1147 | 0.2278 |

## D ALGORITHM

The algorithm of the proposed method is illustrated in 1.

---

**Algorithm 1** The proposed algorithm (FCVL)

---

**Input**: Training data $(x, y)$, detector network $f$ with parameter $\theta$, learning rate $\beta$, probability $p_i$ to do Frequency Jitter , probability $p_f$ to do Amplitude Transfer.
**Output**: The optimized network parameter $\theta^*$.

1: **while** $t \leq T$ **do**
2:      # Fourier-based data augmentation at image level.
3:      Sample $p_0 \sim U(0, 1)$
4:      **for** $(x, y)$ **do**
5:        **if** $p_0 \leq p_i$ **then**
6:          Perform Frequency Jitter according to Eq. 3.
7:          Obtain augmented image $\hat{x}$ according to Eq. 4.
8:        **else**
9:          $\hat{x} \leftarrow x$
10:        **end if**
11:      **end for**
12:      # Fourier-based domain perturbation at feature level.
13:      Sample $p_1 \sim U(0, 1)$
14:      **for** intermediate features $X$ **do**
15:        **if** $p_1 \leq p_f$ **then**
16:          Perform Amplitude Transfer according to Eq.6 - Eq.9.
17:          Obtain perturbed features $\hat{X}$ according to Eq.4.
18:        **else**
19:          $\hat{X} \leftarrow X$
20:        **end if**
21:      **end for**
22:      #Cross-view Semantic Consistency Loss.
23:      $\theta \leftarrow \theta - \beta \cdot \nabla_\theta \mathcal{L}_{\text{train}}((\hat{x}, y), \hat{X}; \theta)$ ;
24: **end while**
25: **return** Save the optimized network $f(\theta^*)$.

---

# E   PROOF

## E.1   PROOF FOR THEOREM1

**Lemma 1.** *Let $G = (\mathcal{X}, W)$ be the augmentation graph, $r$ be the number of underlying classes. There exists an extended labeling function $\hat{y}$ such that*

$$\phi^{\hat{y}} = \sum_{x,x' \in \mathcal{X}} W_{xx'} \cdot \mathbb{I}[\hat{y}(x), \hat{y}(x')] \leq 2\alpha. \tag{19}$$

**Lemma 2.** *(Theorem B.3 (HaoChen et al., 2022)). Assume the set of augmented data $\mathcal{X}$ is finite. Let $f^*$ be the optimal encoder. Then, for any labeling function $\hat{y} : \mathcal{X} \leftarrow [r]$, there exists a linear probe $B^*$ such that*

$$\mathbb{E}_{\overline{x} \sim \mathcal{P}_{x \sim \mathcal{A}(\cdot | \overline{x})}} = [\|y(\overline{x}) - B^* f^*(x)\|_2^2] \leq \frac{\phi^y}{\lambda_{k+1}} + 4\Delta(y, \hat{y}), \tag{20}$$

*where $\lambda_{k+1}$ denotes the $k + 1$-th smallest eigenvalue of the Laplacian matrix $\mathbb{L}$; $\Delta(y, \hat{y})$ denotes the average disagreement between $\hat{y}$ and the ground-truth labeling $y$.*

According to above lemmas, for detection task, the optimal encoder $f^*$, BEV projection module $P_{\text{BEV}}^*$, a learned classification head $C^*$ and regression head $R^*$ on augmented data $\mathcal{X}$, its linear probing error has the following generalization upper bound,

$$\mathcal{E}(f^*, P_{\text{BEV}}^*, C^*, R^*) \leq \frac{\phi^{y_c}}{\lambda_{k+1}} + 4\Delta(y_c, \hat{y_c}) + 4\Delta(y_r, \hat{y_r})$$
$$\leq \frac{2\alpha}{\lambda_{k+1}} + 4\Delta(y_c, \hat{y_c}) + 4\Delta(y_r, \hat{y_r}), \tag{21}$$

where $\alpha$ denotes the labeling error caused by data augmentation; $\Delta(y_c, \hat{y}_c)$ denotes the average disagreement between $\hat{y}_c$ and the ground-truth labeling $y_c$ for classification; $\Delta(y_r, \hat{y}_r)$ denotes the average disagreement between $\hat{y}_r$ and the ground-truth labeling $y_r$ for regression.

### E.2 PROOF FOR THEOREM2

The optimal linear predictor for $y = \mathcal{X}_p \phi + \epsilon$ is

$$\phi^* = \text{argmin}[(y - \mathcal{X}_p \phi)^T (y - \mathcal{X}_p \phi)] \tag{22}$$

$$= (\mathbf{E}[\mathcal{X}_p^T \mathcal{X}_p])^{-1} \mathbf{E}[\mathcal{X}_p^T y] \tag{23}$$

$$= \phi + (\mathbf{E}[\mathcal{X}_p^T \mathcal{X}_p])^{-1} (\mathbf{E}[\mathcal{X}_p^T y] - \mathbf{E}[\mathcal{X}_p^T \mathcal{X}_p] \phi) \tag{24}$$

$$= \phi + (\mathbf{E}[\mathcal{X}_p^T \mathcal{X}_p])^{-1} \mathbf{E}[\mathcal{X}_p^T (y - \phi^T \mathcal{X}_p)] \tag{25}$$

$$= \phi + (\mathbf{E}[\mathcal{X}_p^T \mathcal{X}_p])^{-1} \mathbf{E}[\mathcal{X}_p^T \epsilon] \tag{26}$$

$$= \phi + (\mathbf{E}[\mathcal{X}_p^T \mathcal{X}_p])^{-1} (\mathbf{E}[\mathcal{X}_p^T] \mathbf{E}[\epsilon] + \text{Cov}(\mathcal{X}_p, \epsilon)) \tag{27}$$

$$= \phi \tag{28}$$

If input data $\mathcal{X}$ is deteriorated due to data augmentation in pixel domain, the phase components, $\mathcal{X}_p = \mathcal{X}_{p^-}$, is no longer a distribution with $\mathbf{E}[\mathcal{X}_{p^-}] = 0$. Then, the predictor $\phi$ is biased:

$$\phi^* = \phi + (\mathbf{E}[\mathcal{X}_{p^-}^T \mathcal{X}_{p^-}])^{-1} (\mathbf{E}[\mathcal{X}_{p^-}^T] \mathbf{E}[\epsilon]) \tag{29}$$

## F VISUALIZATION ANALYSIS

### F.1 VISUALIZATION OF PROPOSED FHiAug

In this section, we visualize diverse styles generated via FHiAug. Visualization of Frequency Jitter at image level and visualization of style variations via Amplitude Transfer are shown in Fig.6 and 7. To better illustrate the phase component, we provide more examples of changing phase components in Fig.8.

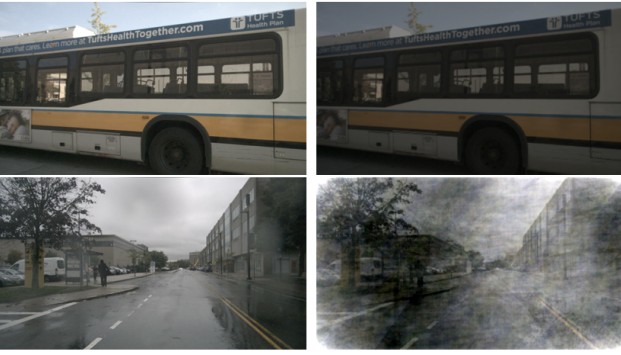

Figure 6: Frequency Jitter at image level. The top line is adjusting the amplitude. It mainly influence the image brightness. The bottom line is adjusting the phase. As it can been, main structures of different targets have been retained.

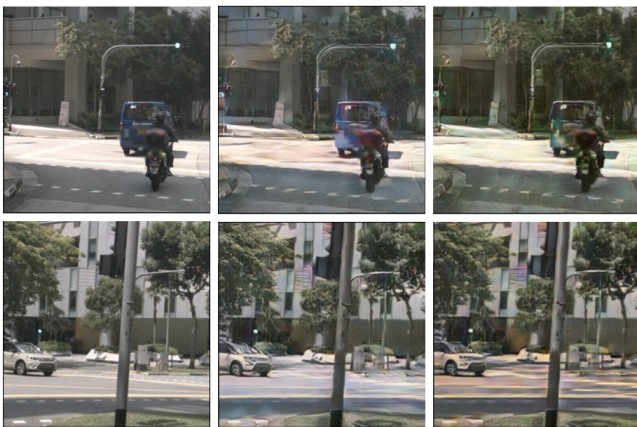

Figure 7: Visualization of style variations via Amplitude Transfer. The left cols are original pictures. The other two cols are styled pictures.

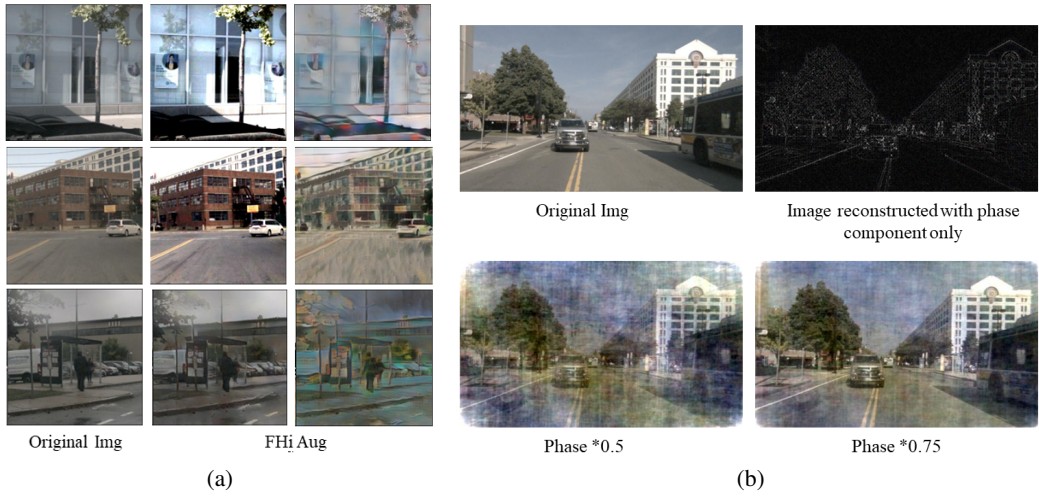

Figure 8: (a) Visualized results of diverse styles generated via FHiAug. For the same image, via FHiAug, we can generate multiple samples, which can facilitate the model to learn more domain invariant feature. (b) The sample on the top right is the image reconstructed with the phase only. As it can be seen, the phase components mainly contain the semantic information. The images in the bottom show how the image changing when adjusting the strength of phase components only. The image after the phase changing is similar to dirty lenses and weather changes in real world, still preserving key BEV prediction information.

## F.2 VISUALIZED DETECTION RESULTS

Notably, FCVL greatly improves the performance for *Snow*. We visualize some detection results of these samples to compare the performance between baseline models and FCVL in Fig.9. Under the condition of *Snow*, baseline model misses detecting the small targets severely, while FCVL can greatly alleviate the problem of missing detection. Compared with CPerb, FCVL still shows more stable and more accurate localization and recognition ability.

Besides, we test our method in the night with the model training on daylight samples only. This example in Fig. 10 well demonstrates that our model can robustly deal with rapid environmental changes, such as variations in lighting conditions.The model is trained on only daylight samples with the proposed FCVL. As it can be seen, in the distance where vehicles are dense and the lighting is very strong, the model can stably detect the targets. As the vehicles move, the lighting becomes

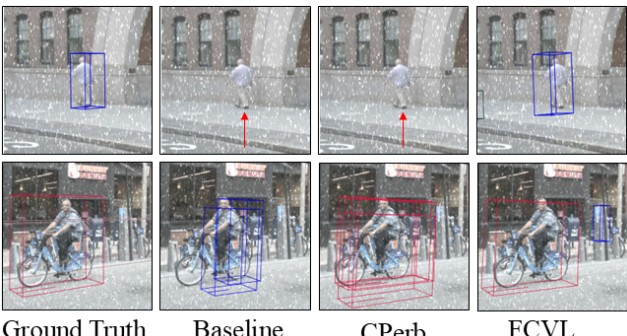

Ground Truth    Baseline    CPerb    FCVL

Figure 9: Visualized detection results of baseline and FCVL from *Snow* set.

normal, and the model detects normally. Although the model has only seen normal daylight samples, with our proposed FCVL, it also performs well under the extreme changes in light condition at night.

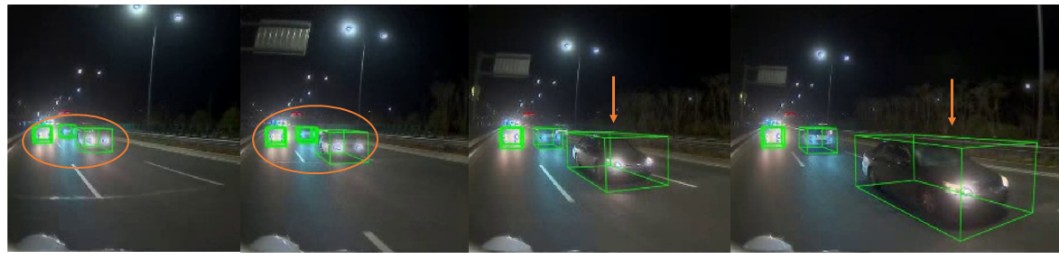

Figure 10: Visualized detection results of FCVL at night with light variations.

