# OpenReview forum: "FCVL: Fourier Cross-View Learning for Generalizable 3D Object Detection in Bird’s Eye View"
_ICLR.cc/2025/Conference — ICLR 2025 Conference Withdrawn Submission_

### Official Review · Reviewer_mXGu · 2024-10-22

**Soundness:** 2
**Presentation:** 2
**Contribution:** 2
**Rating:** 3
**Confidence:** 4

**Summary:**

This paper studies how to boost the generalization of monocular 3D object detectors, like BEVFormer, when only a single domain of data is available. To this end, this paper develops techniques to boost the model domain generalization via augmentation in the frequency domain. A theoretical analysis based on the augmentation graph theory is provided. Extensive experiments on the nuScenes benchmark are conducted to verify the effectiveness of the proposed techniques.

**Strengths:**

1. The pipeline figure is well designed.
2. In experiments, this work compares its own method with many other counterparts.

**Weaknesses:**

1. The studied problem, improving model domain generalization with only a single domain of data, is not very meaningful. First, driving data is not so expensive to collect that only a single domain of data can be collected. For companies, diverse domains of data are absolutely available. I cannot think out a scenario that only a single domain of data can be used in practical applications.
2. The presentation needs significant improvement. For example, the story flow of Intro is not fluent.
3. The technique contributions, data augmentations in the frequency domain, are not very interesting. This is a somewhat straightforward idea. there are many attempts that design operations in the frequency domain, although maybe not in 3D object detection. However, these frequency-domain based methods are not really widely adopted after many years of research, and this work does not present a significant difference from them.
4. The theoretical analysis seems not to add valuable information to this paper. Do not use mathematical formulations just because want to add mathematical formulations. This will make the paper more difficult to understand.

**Questions:**

See the weakness. The authors can remind me if I overlook or misunderstand something important.

---

> ### Author Response · Authors · 2024-11-25
>
> We would like to thank the reviewer for taking the time to read our submission and for valuable feedback concerning our paper.
>
> “Summary: This paper studies how to boost the generalization of **monocular 3D object detectors, like BEVFormer**, when only a single domain of data is available. ”
>
> **Firstly, we kindly remind the reviewer that it is not monocular 3D object detection.**  It is a multi-view 3D object detection approach, which is often referred to as the BEV object detections in autonomous driving. This is indeed a very different area: **the same target object could appear in adjacent perspectives, resulting in cross-view situation** (please kindly check Figure 3 in the paper) and one of our main contributions is leveraging the natural cross-view features from multiple inputs to enhance generalization, utilizing its multi-view nature.
>
>
> **W1**
>
> Although companies may collect data from multiple domains, it is still meaningful to study how to improve the generalization ability of models using only data from a single domain.  For research purposes, we are interested in how to approaching human-level generalization abilities, which means **you do not need the whole world's data for driving**. Indeed, humans do not have to experience every domain (daytime, night, rainy, cloudy, and snowy) to learn to drive, which is in stark contrast with the common practice of collecting an enormous amount of data.
>
> Furthermore, for democratic purposes, we may not wish the autonomous driving to be solely controlled in a small group. We wish with the development of the generalization research, more people including resource-limited non-profit academic institutions can produce generalizable and safe autonomous driving systems which could benefit every one.  **Besides, in certain situations, such as rapid deployment in emergencies or specific fields, there might indeed be only data from a single domain available.**  For example, if we would like to quickly develop an autocar for earthquake rescues, we cannot expect too many data in this kind of domains. This further justifies the necessities of this research direction.
>
> How to improve the generalization ability of models with limited data in situations where resources are constrained or specific domain data is difficult to obtain is a worthwhile research question.  Focusing on single-domain generalization (SDG) not only addresses practical constraints but also provides a more robust evaluation of model adaptability.
>
> **W2**
>
> Thank you very much for your comments. We will improve the fluency of this paper in later versions.
>
> **W3**
>
> Thank you for the constructive comment. Frequency-domain based methods and their variants are actually widely in commercial autonomous driving vehicles especially when the computation power  is extremely limited. For example, in the image processing ISP pipelines, they can be used for denoising the analog signals. We hope our method could potentially be widely adopted in the future.
>
> **W4**
>
> Thank you for the feedback. Using mathematical formulations here is to help providing further theoretical understanding of the working mechanisms for this method. This can provide readers more information in addition to the empirical results present in the paper.
>
> We thank again for the reviewer's helpful suggestions.

---

> > ### Comment · Reviewer_mXGu · 2024-11-25
> >
> > Thanks for the careful reply.
> >
> > 1. The multi-view detection in nuScenes is also monocular. There is no significant visual overlap between different views. Do not worry that I will not misunderstand such simple concepts. I have worked on this topic for years and published may papers.
> > 2. I agree that the resource of a research lab is limited. But if a problem can be addressed easily be data cheaply, its practical meaning for using algorithm is limited. Selecting a valuable study topic is important, as the lab resource is very precious.
> > 3. For the fluency and mathematical formulation issues, I just provide my justifications. I have checked the paper again and still hold my opinions. Wish that they are benefitial to improving your paper.

---

> ### Author Response · Authors · 2024-11-25
>
> We have updated the reply above, which should also have addressed your comments. Please checked the updated version.
>
> We would like to emphasize the significance of improving model generalization with single domain data  once again.
>
> 1. Generalization ability refers to how well a model performs in the scenes it has not seen before. Since they are **"unseen scenes“**,  this is hard to be solved by collecting more data alone. Data is infinite, and it is impossible to collect data on all possible scenarios.
>
> 2. The setting of single domain generalization is a realistic yet more challenging scenario. As deep neural networks are very good at memorizing all the training data, **single domain generalization can truly reflect an algorithm's generalization ability in real-world environments.** Humans beings do not have to be taught on all luminace and weather conditions to learn to drive.
>
> 3.  For multi-view 3D detection task , the collection of data and annotating 3D bounding boxes is much more time-consuming and labor-intensive than  common 2D classification tasks.  There is no dispute about this point. Studying the single-domain generalization of multi-view 3D detection , (1) reduces the dependence on more annotated data, (2) allows the model to better cope with complex and variable driving environments, which is of great significance for the safety and reliability of autonomous driving.

---

### Official Review · Reviewer_fBAM · 2024-10-31

**Soundness:** 3
**Presentation:** 3
**Contribution:** 3
**Rating:** 6
**Confidence:** 4

**Summary:**

The paper introduced a interesting problem, that is 3D object detection models trained on a single source domain can be generalized to others. The authors proposed FCVL (Fourier Cross-View Learning), a framework to improve the generalization of Bird's Eye View (BEV) 3D object detection models when trained on a single source domain. The key contributions include:

- A Fourier Hierarchical Augmentation (FHiAug) strategy that works at both image and feature levels to increase domain diversity without requiring additional modules.
- A Fourier Cross-View Semantic Consistency Loss that leverages natural multi-view inputs to learn domain-invariant features.
- Theoretical guarantees for the effectiveness of FHiAug using augmentation graph theory.
- Extensive experimental validation showing superior performance compared to existing domain generalization methods across various test domains.

**Strengths:**

The proposed method seems novel. They innovatively use of Fourier domain for both augmentation and cross-view consistency with non-parametric approach that doesn't require additional training modules. Meanwhile, The authors provided solid theoretical guarantees through augmentation graph theory and clear mathematical formulation and proofs for the proposed methods. Extensive experiments across multiple frameworks (BEVFormer, BEVDepth, BEVDet) are conducted.

**Weaknesses:**

- The proposed method has seveal hyperparameters to tune, in the paper, the authors did not specifically point it out how to make it work. If the authors could elaborate how to set up these hyperparameters and how do they affect the final performance, it will be better.
- The authors could discuss more on the failure cases or limitations
- The experimental results are preseneted mainly on synthetic corruptions (nuScenes-C), how about others datasets in the self-driving field? It will be beneficial to see more diverse real-world testing scenarios

**Questions:**

Please refer to weaknesses part for the questions.

---

> ### Author Response · Authors · 2024-11-24
>
> We would like to thank the reviewer for taking the time to read our submission and for valuable feedback concerning our paper.
>
> **W1 The proposed method has seveal hyperparameters to tune, in the paper, the authors did not specifically point it out how to make it work. If the authors could elaborate how to set up these hyperparameters and how do they affect the final performance, it will be better.**
>
> Thank you for the constructive suggestion. These hyperparameters mainly consist of two parts: the probability of augmentation and the intensity of augmentation (more details can be found in the methodology (Sec2.2,2.3) and hyperparameter analysis (Sec4.3)). As the probability $p$  and the intensity $\alpha$ increase, this method can create more samples with more diverse styles to improve the generalization. As is shown in the Fig.4(c) in hyperparameters analysis: initially, as the probability $p$  and the intensity $\alpha$ increase, the out-of-domain performance gradually improves. After reaching a certain level of probability and intensity, further changes in the parameters will no longer cause drastic changes in OOD performances, indicating that the model is stable against hyper-parameter misspecifications as long as the hyper-parameters are within reasonable ranges.
>
> **W2 The authors could discuss more on the failure cases or limitations**
>
> We have added subsection Limitations in the revised paper.
>
> Currently, this method involves several hyperparameters that require fine-tuning. In our future work, we aim to explore supplementary techniques to minimize the time spent on hyperparameter optimization and to further augment the performance of FCVL.  For snowy weather, we have already improved by 10 points, but the performance in snowy conditions is still much worse compared to the performance in other scenarios such as low light.  Consequently, there is a substantial potential for enhancement in adverse weather conditions.
>
> **W3 additional evaluation results using different datasets**
>
> Thank you for the constructive suggestion. Following this suggestion, we have introduced a new state-of-the-art (SOTA) baseline called Far3D and conducted thorough comparative experiments on the updated dataset, Argoverse 2. Additional results across more datasets have been included in Table 2 of the revised paper. On Argoverse 2, our method still has a significant advantage over other domain generalization methods. More visualized results are shown in the anonymous link.
> https://drive.google.com/file/d/1p7ATTCM55GQNH7JltW6dJsBF3NWJaNXi/view?usp=drive_link
>
>
> Secondly, we have assessed our method's performances in real-world scenarios, particularly in conditions with abrupt light changes. Our method was tested at night using a model trained solely on daylight samples. We have visualized several detection results from image sequences captured in real-world environments and provided an analysis on pages 19-20 in Appendix F.2 (Lines 1022-1050) of the revised paper.  We have also uploaded the visualization results in the anonymous link. https://drive.google.com/file/d/15vYYmeviYDbLy9ugJOsfQ55z5XB3QBS2/view. Figure 10 illustrates a compelling example of our model's ability to robustly handle rapid environmental changes, such as fluctuations in lighting conditions. The model, trained exclusively on daylight samples with the proposed FCVL, demonstrates consistent performances in areas with dense vehicle traffic and intense lighting, effectively detecting targets under these challenging conditions. As the vehicles transition into areas with normalized lighting, the model's detection capabilities return to standard operation. Notably, despite being exposed only to typical daylight samples, the model, augmented with our proposed FCVL, performs significantly better even under the extreme lighting variations encountered at night.
>
> Thank you again for your time and suggestions. Please do not hesitate to contact us if you have any further questions.

---

### Official Review · Reviewer_p6Tr · 2024-11-02

**Soundness:** 2
**Presentation:** 3
**Contribution:** 2
**Rating:** 5
**Confidence:** 5

**Summary:**

This paper aims to address the challenge of Single Domain Generalization in BEV-based multi-camera 3D object detection and proposes a Fourier Cross-View Learning (FCVL) framework. This framework consists of a non-parametric Fourier Hierarchical Augmentation (FHiAug) at both image and feature levels to enhance data diversity and a Fourier Cross-View Semantic Consistency Loss to facilitate model to learn more domain-invariant features from adjacent perspectives. Additionally, they provide theoretical guarantees through augmentation graph theory.

**Strengths:**

1. This paper is clearly structured and written-well. Proper formulization makes the processes of the method easy to understand.
2. This paper focuses on an interesting issue, namely that Single Domain Generalization in BEV-based multi-camera 3D object detection.
3. The paper offers extensive experiments and rigorous theoretical guarantees.

**Weaknesses:**

1. The Fourier Cross-View Semantic Consistency Loss constructs positive and negative samples by splitting adjacent perspectives into halves. However, this approach has a limitation: each segment contains not only foreground objects but also complex background interference, which requires further analysis.
2. This paper focuses on addressing the domain generalization problem but currently only provides numerical experiments to demonstrate the method's effectiveness. It is recommended to supplement the analysis with commonly used feature distribution visualizations in the field to provide a more intuitive demonstration of the proposed method's efficacy.

**Questions:**

1. Please refer to the Paper Weaknesses mentioned above.
2. How is the number of positive and negative samples set in the Fourier Cross-View Semantic Consistency Loss?
3. Most of the current experiments are based on relatively older detectors. Is there still a significant performance improvement when applied to the latest SOTA BEV-based detectors?

---

> ### Author Response · Authors · 2024-11-24
>
> We would like to thank the reviewer for taking the time to read our submission and for valuable feedback concerning our paper.
>
> **W1 The Fourier Cross-View Semantic Consistency Loss constructs positive and negative samples by splitting adjacent perspectives into halves. However, this approach has a limitation: each segment contains not only foreground objects but also complex background interference, which requires further analysis.**
>
> This is indeed a valuable suggestion. In our experiments, we find that the target objects across different viewpoints occupy most of the foreground area (as we have split each perspective into halves). We conducted a separate experimental analysis of the Cross-View Loss. Under the current experimental setup, our cross-view learning loss has achieved good experimental results (Table 9).  We have visualized more cross-view samples with GradCam, it can be seen that the same object from different perspectives has a stronger feature response compared with background. (https://drive.google.com/file/d/155vOi-sRGqk4P-DIYWJbXhgMLNiEKjTH/view?usp=drive_link)
>
> Such cross-view targets
> are common in multi-camera inputs, providing natural opportunities to observe the same object from different perspectives. To exploit this, we propose the
> Fourier Cross-View Semantic Consistency Loss to help
> the model learn more domain-invariant features from adjacent
> views.
>
> **W2 supplement the analysis with commonly used feature distribution visualizations**
>
> Thank you for your constructive comment.  We employ t-SNE for visualizing the Bird's Eye View (BEV) features across various domains, with the results presented in Section 4.5 of our paper. We have also uploaded the visualization results in the anonymous link. https://drive.google.com/file/d/199U0ufsXO0QeGGVWu9K4Vv2uxUaLR3fG/view?usp=drive_link. The visualization reveals that the features extracted from BEVDet for different domains are not only distant from one another but also loosely dispersed throughout the feature space. In contrast, after applying our FCVL optimization, the features from the four domains are more tightly clustered and interconnected, aligning with the principles of augmentation graph theory. FCVL enhances the connectivity of the augmentation graph between the source and unseen domains, thereby significantly bolstering the model's generalization capabilities.
>
> **Q2 How is the number of positive and negative samples set in the Fourier Cross-View Semantic Consistency Loss?**
>
> The ratio of positive to negative sample pairs is 1:10.
>
> **Q3 additional evaluation results using different datasets**
>
> Thank you for the constructive suggestion. Following this suggestion, we have introduced a new state-of-the-art (SOTA) baseline called Far3D and conducted thorough comparative experiments on the updated dataset, Argoverse 2. Additional results across more datasets have been included in Table 2 of the revised paper. On Argoverse 2, our method still has a significant advantage over other domain generalization methods. More visualized results are shown in the anonymous link.
> https://drive.google.com/file/d/1p7ATTCM55GQNH7JltW6dJsBF3NWJaNXi/view?usp=drive_link
>
> Secondly, we have assessed our method's performances in real-world scenarios, particularly in conditions with abrupt light changes. Our method was tested at night using a model trained solely on daylight samples. We have visualized several detection results from image sequences captured in real-world environments and provided an analysis on pages 19-20 in Appendix F.2 (Lines 1022-1050) of the revised paper. We have also uploaded the visualization results in the anonymous link. https://drive.google.com/file/d/15vYYmeviYDbLy9ugJOsfQ55z5XB3QBS2/view.  Figure 10 illustrates a compelling example of our model's ability to robustly handle rapid environmental changes, such as fluctuations in lighting conditions. The model, trained exclusively on daylight samples with the proposed FCVL, demonstrates consistent performances in areas with dense vehicle traffic and intense lighting, effectively detecting targets under these challenging conditions. As the vehicles transition into areas with normalized lighting, the model's detection capabilities return to standard operation. Notably, despite being exposed only to typical daylight samples, the model, augmented with our proposed FCVL, performs significantly better even under the extreme lighting variations encountered at night.
>
> Thank you again for your time and suggestions. Please do not hesitate to contact us if you have any further questions.

---

> > ### Comment · Reviewer_p6Tr · 2024-11-25
> > **There are still some concerns that have not been well addressed.**
> >
> > Thanks for the rebuttal. I have carefully read the author's response and the comments of other reviewers, but there are still some concerns that have not been well addressed. 1. The Fourier Cross-View Semantic Consistency Loss constructs positive and negative samples by splitting adjacent perspectives into halves. However, this approach has a limitation: each segment contains not only foreground objects but also complex background interference, which remains unaddressed. 2. I tend to agree with the concern raised by reviewer mXGu: the primary purpose of multi-view settings is to cover the full surround view, but the overlapping areas between different views are relatively limited. 3. I tend to agree with the concern raised by reviewer mXGu: the studied problem—improving model domain generalization using only a single domain of data—lacks substantial significance. Based on the above reasons, I think the authors may will further refine the method and I will lower my score.

---

> ### Author Response · Authors · 2024-11-25
>
> As for 1& 2,  the phenomenon of objects crossing viewpoints exist naturally and very common. We propose to utilize this point and improve the ood performance. In our experiments, we find that the target objects across different viewpoints occupy most of the foreground area (as we have split each perspective into halves). We conducted a separate experimental analysis of the Cross-View Loss. Under the current experimental setup, our cross-view learning loss has achieved good experimental results (Table 9). We have visualized more cross-view samples with GradCam, it can be seen that the same object from different perspectives has a stronger feature response compared with background. (https://drive.google.com/file/d/155vOi-sRGqk4P-DIYWJbXhgMLNiEKjTH/view?usp=drive_link) . Such cross-view targets
> are common in multi-camera inputs, providing natural opportunities to observe the same object from different perspectives. To exploit this, we propose the
> Fourier Cross-View Semantic Consistency Loss to help
> the model learn more domain-invariant features from adjacent
> views.
>
> The main advantage of surround-view input is its ability to provide more comprehensive environmental information, which is very helpful for detecting and tracking objects across different views.  There are also other works may not about generalization studies but related to cross-view features. This work[1] is about generating surround-view data and they design “a cross-view attention module, ensuring consistency across multiple camera views”.  These can verify that the phenomenon of objects crossing viewpoints exist naturally and very common.
>
> [1] Gao, Ruiyuan, et al. "Magicdrive: Street view generation with diverse 3d geometry control.” ICLR,2024
>
> As for 3,  we would like to emphasize the significance of improving model generalization with single domain data  once again.
>
> We would like to emphasize the significance of improving model generalization with single domain data  once again.
>
> 1. Generalization ability refers to how well a model performs in the scenes it has not seen before. Since they are **"unseen scenes“**,  this is hard to be solved by collecting more data alone. Data is infinite, and it is impossible to collect data on all possible scenarios.
>
> 2. The setting of single domain generalization is a realistic yet more challenging scenario. As deep neural networks are very good at memorizing all the training data, **single domain generalization can truly reflect an algorithm's generalization ability in real-world environments.** Humans beings do not have to be taught on all luminace and weather conditions to learn to drive.
>
> 3.  For multi-view 3D detection task , the collection of data and annotating 3D bounding boxes is much more time-consuming and labor-intensive than  common 2D classification tasks.  There is no dispute about this point. Studying the single-domain generalization of multi-view 3D detection , (1) reduces the dependence on more annotated data, (2) allows the model to better cope with complex and variable driving environments, which is of great significance for the safety and reliability of autonomous driving.

---

### Official Review · Reviewer_K6Mg · 2024-11-03

**Soundness:** 3
**Presentation:** 2
**Contribution:** 2
**Rating:** 5
**Confidence:** 3

**Summary:**

This paper addresses the challenge of improving the generalization of BEV detection models for autonomous driving. The authors introduce a novel framework called Fourier Cross-View Learning, which includes two key components:
1. Fourier Hierarchical Augmentation, an augmentation strategy that operates in the frequency domain to enhance domain diversity.
2. Fourier Cross-View Semantic Consistency Loss, which helps the model learn domain-invariant features.

**Strengths:**

1. The paper introduces the Fourier Cross-View Learning framework, featuring Fourier Hierarchical Augmentation to enhance domain diversity without additional complexity.
2. The paper proposes fourier cross-view semantic consistency loss to improving generalization in real-world scenarios.

**Weaknesses:**

1. Since the primary augmentation occurs on 2D images, it can be difficult to differentiate it from standard 2D augmentation techniques. A more detailed explanation of these methods might enhance clarity for readers less familiar with them.
2. Because the evaluation is solely conducted on the nuScenes-C dataset, there is a risk that the models may be trained to excel in this specific context, potentially neglecting more challenging real-world scenarios. Testing exclusively on one dataset limits the robustness of the findings and reduces the overall credibility of the conclusions.

**Questions:**

Could you provide additional evaluation results using different datasets? Relying solely on the nuScenes-C dataset increases the risk of overfitting through parameter tuning.

---

> ### Author Response · Authors · 2024-11-24
>
> We would like to thank the reviewer for taking the time to read our submission and for valuable feedback concerning our paper.
>
> **W1 Since the primary augmentation occurs on 2D images, it can be difficult to differentiate it from standard 2D augmentation techniques. A more detailed explanation of these methods might enhance clarity for readers less familiar with them.**
>
> Thank you for the valuable suggestion. In the introduction, we have outlined the challenges associated with directly applying standard 2D augmentation techniques—such as geometric transformations, style transfer, and data generation to the Bird's Eye View (BEV) task. Following your guidance for better clarity, we have included a more comprehensive overview of these 2D methods in Appendix A of the revised manuscript, owing to spatial constraints within the main text. Additionally, we have elaborated on the distinctions between our approach and other frequency-domain methodologies in Appendix A. This indeed improves the clarity of our paper.
>
> **W2 & Q2 additional evaluation results using different datasets**
>
> Thank you for the constructive suggestion. Following this suggestion, we have introduced a new state-of-the-art (SOTA) baseline called Far3D and conducted thorough comparative experiments on the updated dataset, Argoverse 2. Additional results across more datasets have been included in Table 2 of the revised paper. On Argoverse 2, our method still has a significant advantage over other domain generalization methods. More visualized results are shown in the anonymous link.
> https://drive.google.com/file/d/1p7ATTCM55GQNH7JltW6dJsBF3NWJaNXi/view?usp=drive_link
>
> Secondly, we have assessed our method's performances in real-world scenarios, particularly in conditions with abrupt light changes. Our method was tested at night using a model trained solely on daylight samples. We have visualized several detection results from image sequences captured in real-world environments and provided an analysis on pages 19-20 in Appendix F.2 (Lines 1022-1050) of the revised paper. We have also uploaded the visualization results in the anonymous link. https://drive.google.com/file/d/15vYYmeviYDbLy9ugJOsfQ55z5XB3QBS2/view.  Figure 10 illustrates a compelling example of our model's ability to robustly handle rapid environmental changes, such as fluctuations in lighting conditions. The model, trained exclusively on daylight samples with the proposed FCVL, demonstrates consistent performances in areas with dense vehicle traffic and intense lighting, effectively detecting targets under these challenging conditions. As the vehicles transition into areas with normalized lighting, the model's detection capabilities return to standard operation. Notably, despite being exposed only to typical daylight samples, the model, augmented with our proposed FCVL, performs significantly better even under the extreme lighting variations encountered at night.
>
> Thank you again for your time and suggestions. Please do not hesitate to contact us if you have any further questions.

---

> ### Comment · Reviewer_K6Mg · 2024-12-03
>
> Thank you for your thoughtful response, which addressed most of my concerns. After carefully reviewing the comments from the other reviewers, I still have some concerns that need to be addressed. While the rebuttal argues that single-domain generalization accurately reflects an algorithm's generalization ability in real-world environments, I believe this should be validated through experiments rather than relying solely on theoretical reasoning.

---

> > ### Author Response · Authors · 2024-12-03
> >
> > Thank you for your reply. In the real world, humans beings do not have to be taught on all light or weather conditions to learn to drive. While existing neural networks, under the condition of single domain and limited data, cannot possess generalization capabilities like humans. **The numerous experimental results  in the paper** indicate that baseline models(exisiting 3D detectors) trained on single-domain data experienced a significant decline in performance when tested on eight other unseen domains. Our algorithm aims to approach human-level generalization ability in such a setting.
> >
> >  Single-Domain Generalization (SDG) [1-3] involves training on a single source domain with the goal of generalizing to multiple unseen target domains. The challenge in SDG is that the model must capture sufficient generalizable features from a single training distribution to perform well on different test distributions. Common Domain Generalization (DG)[4] typically involves training with data from multiple source domains, aiming to learn a model that can generalize to unknown target domains. Unlike SDG, DG has access to data from multiple source domains during training. In comparison, SDG is more challenging.
> >
> >
> > [1]Yuan, Junkun, et al. "Domain-specific bias filtering for single labeled domain generalization." International Journal of Computer Vision 131.2 (2023): 552-571.
> >
> > [2] Zheng, Guangtao, Mengdi Huai, and Aidong Zhang. "AdvST: Revisiting Data Augmentations for Single Domain Generalization." Proceedings of the AAAI Conference on Artificial Intelligence. Vol. 38. No. 19. 2024.
> >
> > [3]Vidit, Vidit, Martin Engilberge, and Mathieu Salzmann. "Clip the gap: A single domain generalization approach for object detection." Proceedings of the IEEE/CVF conference on computer vision and pattern recognition. 2023.
> >
> > [4] Zhou, Kaiyang, et al. "Domain generalization: A survey." IEEE Transactions on Pattern Analysis and Machine Intelligence 45.4 (2022): 4396-4415.

---

### Official Review · Reviewer_g8Uo · 2024-11-04

**Soundness:** 3
**Presentation:** 3
**Contribution:** 3
**Rating:** 5
**Confidence:** 4

**Summary:**

This paper introduces FCVL (Fourier Cross-View Learning), a framework to improve the generalization capability of Bird's Eye View (BEV) 3D object detection models when trained on single-source data.

**Strengths:**

1. While most previous work focuses on multi-domain generalization, this paper tackles the more challenging and practical problem of single-domain generalization. This is especially relevant for autonomous driving where collecting multi-domain data is expensive and time-consuming.
2. The paper provides a formal analysis using augmentation graph theory, connecting practical augmentations to theoretical guarantees. The theoretical analysis provides insights into why the method works.

**Weaknesses:**

1. Fourier transformations are computationally intensive and scale quadratically with image dimensions. Lack of analysis of how the method scales with increasing image resolution or number of cameras.
2. Limited discussion of computational overhead and training time, which is critical for practical implementation.  How does the computational complexity of FCVL compare to baseline methods, particularly during training and inference? e.g. total training time comparison with baselines, additional memory requirements and computations for frequency domain operations.
3. In table.1, it seems FCVL didn't show superior performance on the normal validation set.
4. Missing references:
- HotBEV: Hardware-oriented transformer-based multi-view 3D detector for BEV perception
- Bevformer v2: Adapting modern image backbones to bird's-eye-view recognition via perspective supervision
- Clip-bevformer: Enhancing multi-view image-based bev detector with ground truth flow
- Ocbev: Object-centric bev transformer for multi-view 3d object detection
- BEVNeXt: Reviving Dense BEV Frameworks for 3D Object Detection

**Questions:**

1. What is the rationale behind the specific choices of frequency domain transformations? Were other alternatives considered?
2. Does the method maintain its effectiveness when dealing with rapid environmental changes (e.g., entering/exiting tunnels)?

---

> ### Author Response · Authors · 2024-11-24
>
> We would like to thank the reviewer for taking the time to read our submission and for valuable feedback concerning our paper.
>
> **W1&W2 training and inference efficiency**
>
> Thank you for the valuable question. We add a new subsection "Efficiency Analysis" (Sec4.4, Page 9) in the revised paper. In this part, we make more analysis to delve into the proposed FCVL. We investigate how the method scales with increasing image resolution and computational complexity for practical implementation. The results are listed in Table 5 of the revised paper. As it can been seen, (1) at larger image scales, FCVL can still significantly improve the model's generalization performance. (2) There will be a slight increase in training time (+0.11s per training step) during the training phase. However, as the FCVL is only used during the training phase, it introduces no latency in inference phase. Without the need for more time-consuming and costly data collections, the FCVL can improve the generalization performance almost for free. (3) Besides, NVIDIA provides a library called cuFFT to accelerate the Fourier Transform. Many hardware vendors also provide Fourier transformation accelerations. Thus, the speed of our method can be further improved in the training phase.
>
> **W3 In table.1, it seems FCVL didn't show superior performance on the normal validation set.**
>
> Thank you for the constructive feedback. There are already many excellent works on improving the in-distribution performances, mostly from a neural architectural approach. Previous works mainly focus on improving the network’s capacity to improve the performances on the normal validation set, i.e. the in-distribution evaluation set.
>
> In this paper, we provide a novel perspective on improving the performances on the data distributions unseen in the training phase. This arguably provides a more challenging but practical setting for BEV object detection methods evaluations. This is because we cannot collect all data in the world including all colors, shapes, combinations of objects and backgrounds images on the road. As shown in the paper, we achieved competitive performances on the normal validation set, which demonstrates that our method is in par with the baseline methods. We also achieved significant performances on the OOD set (increased by 5-6 \%), this indicates that our method can significantly improve existing BEV object detectors’ generalization performances. Our approach is indeed orthogonal to previous approach which mostly focus on the neural network architectures.
>
>
> **W4 Missing references**
>
> Thank you for the reminder. The references have already been supplemented in the revised paper.

---

> ### Author Response · Authors · 2024-11-27
>
> **Q1 What is the rationale behind the specific choices of frequency domain transformations? Were other alternatives considered?**
>
> Thank you for the insightful question. We opt to operate in the frequency domain as it enables us to distinctly separate phase components, which include semantics and causal cues, from amplitude components, which encompass styles and non-causal cues. Causal cues are pivotal for bolstering the model's resilience to variations across disparate domains. While previous works propose methods to extract causal features or decouple them through sophisticated network architectures, these works focus on the image classification cases. For example, CIRL [1] creates augmented images through a causal intervention module that targets non-causal factors, and AGFA [2] employs adversarial training between a classifier and an amplitude generator to produce a challenging domain for model adaptation. We have tried to adapt these methods on the BEV object detections but they do not show good performances in the experiments. The reason is that for image classification tasks, the key information for classification is often concentrated in the central part of the image. Thus, it is closer to the ideal theoretical settings. However, the BEV object detection tasks are much more complex. Most of parts of the images are backgrounds. Our approach stands out for its stability and efficacy without the need for additional module design or specialized training regimens. In contrast to these techniques, our method offers a straightforward, plug-and-play solution that yields superior generalization outcomes with greater efficiency, particularly beneficial given the intricate nature of BEV-based 3D object detection models. As highlighted in our efficiency analysis, our method is utilized exclusively during the training phase and is not deployed during the inference phase, thus incurring no extra computational overhead for real-world applications.
>
> [1].Lv, Fangrui, et al. “Causality inspired representation learning for domain generalization.” Proceedings of the IEEE/CVF conference on computer vision and pattern recognition. 2022.
>
> [2] Kim, Minyoung, Da Li, and Timothy Hospedales. “Domain generalization via domain adaptation: An adversarial Fourier amplitude approach.” arXiv preprint arXiv:2302.12047 (2023).
>
> **Q2 Does the method maintain its effectiveness when dealing with rapid environmental changes (e.g., entering/exiting tunnels)?**
>
> Thank you for the valuable suggestion. We apologize that we were unable to acquire data for scenarios involving entering and exiting tunnels in the limited time frame. However, we have conducted tests on similar scenarios characterized by abrupt luminance condition changes. We have visualized certain detection results from image sequences captured in real-world settings and provided a detailed analysis on pages 19-20 of the revised paper, specifically in Appendix F.2 (Lines 1022-1050). We have also uploaded the visualization results in the anonymous link.
> https://drive.google.com/file/d/15vYYmeviYDbLy9ugJOsfQ55z5XB3QBS2/view. Our method was tested under nighttime conditions, despite the model being trained exclusively on daylight samples. Figure 10 illustrates a compelling example of our model's ability to robustly handle rapid environmental changes, such as fluctuations in lighting conditions. **Trained solely on daylight samples** with the proposed FCVL, **the model demonstrates consistent performance in dense vehicle areas with intense lighting, successfully detecting targets even under these challenging conditions**. As the vehicles move into areas with normalized lighting, the model's detection capabilities return to normal. Notably, despite being trained only on typical daylight samples, **our model, augmented with FCVL, performs better even under the extreme lighting variations encountered at nights.**
>
> Thank you again for your time and considerations. Please do not hesitate to contact us if you have any further questions.

---

### Author Response · Authors · 2024-11-24
**Global Response**

We sincerely thank all reviewers for providing all the constructive feedback. In terms of Soundness, Presentation, and Contribution, the majority of the reviewers deemed our work to be commendable. We are deeply appreciative of the affirmation received from the diligent and responsible reviewers. We have thoroughly addressed all the concerns raised by the reviewers. A revised version of our paper has been uploaded, with the modifications highlighted in brown.

The primary concerns raised by the reviewers encompass two areas: experimentation on new datasets and computational complexity analysis.

1. We have conducted experiments on the new dataset, Argoverse 2, using a new state-of-the-art (SOTA) detector as the baseline and performed comprehensive comparative experiments (as shown in Table 2) to evaluate our method thoroughly. On Argoverse 2, our method still has a significant advantage over other domain generalization methods.  Additionally, we have assessed our method in real-world scenarios, including abrupt light changes. The visualized results and analysis (Lines 1022-1050) can be found in Appendix F.2 on Page 19 of the revised paper. With the proposed FCVL, the model can perform stably during drastic light changes at night.

2. We have included Efficiency Analysis as a new subsection (Sec 4.4).  In the inference stage, the proposed FCVL enhances the algorithm's generalization performance without increasing the time consumption, which is beneficial for practical applications.

3. We have incorporated additional visualization analysis using t-SNE in a new subsection (Sec 4.5).  Via visualization analysis using t-SNE, we can find that the features of different domains extracted from baseline model are distant from each other and loosely distributed in the feature space. While, after optimization with FCVL, the distribution of different domains becomes more compact and connected, which is in line with augmentation graph theory. FCVL increases the augmentation graph connectivity between source and unseen domains and improve the generalization ability.

If you have any further inquiries, please do not hesitate to post. We thank all reviewers again for all the time and efforts dedicating to our paper.

---

### Note · Authors · 2025-01-23

I have read and agree with the venue's withdrawal policy on behalf of myself and my co-authors.